



# 1 GAIA 5.0 — A five-dimensional geometry for the 3D
# 2 visualization of Earth' climate complexity

Renate C.-Z.-Quehenberger,[1] Sergio dC. Rubin[2] Leyla Kern[3] and Daniel Tirelli[4]
[1] SciArt Research & Institute for a Global Sustainable Information Society (GSIS), Vienna, 1130, Austria
[2] Georges Lemaître Centre for Earth and Climate Research, Earth and Life Institute, Université Catholique de
Louvain, Louvain-la-Neuve, 1348, Belgium
[3] High-Performance Computing Center Stuttgart (HLRS) of the University of Stuttgart, 70569, Germany
[4] The Joint Research Centre (JRC) of the European Commission, Ispra, 21027, Italy
*Correspondence to*: Renate C.-Z.-Quehenberger (rczq@qc-l.eu)
**Abstract.** The artwork presented here is motivated by the wish to communicate an assumed underlying higher-
dimensional space grid for the Universe. It was presented in the digital 3D animated opera named *GAIA 5.0. — A*
*Holographic Image Ambience,* a 'big data' visualisation project that proposes a 5-dimensional (suffix 5.0) geomet-
ric visualization of the Earth from the Gaia hypothesis standpoint: the Earth as a living, thus complex system.
This quasi-crystallography visualisation, situates the Earth in the center of a dynamic 5-dimensional subscendent
space configuration and draws parallels between 19th ether concept (cf. Lord Kelvin, V. Bjerknes a.o.) and the
newly discovered 3D representation of the *Penrose Kites & Darts* tiling (Epitahedra, E±) which allows us to re-
build the presocratian 'living pattern of the world' as 5-dimensional space conforming the Poincaré's dodecahedral
space. -- This allows us to regard the Earth not as big data generator but its underlying space which generates
symmetries in compounds of multi-dimensional spaces with create particles, matter, fluxes, cycles, metabolism,
and meteorological and climate turbulences. Meteorologic and climatic features like the Coriolis effect, Ekman
pumping and the Lorenz attractor are discussed in this geometrical framework from an artistic visualization point
of view. We conclude that those dynamical physical phenomena on Earth can be related to intrinsic geometrical
features of higher dimensional spaces proper to living systems. Finally, a gender perspective and the role of arts as
tool for communication in scientific research of complex systems are briefly discussed.
**Keywords:** Gaia hypothesis; Meteorology; Big Data visualization; Hyper-Euclidean Geometry; 3D representation
of the Penrose kites & darts tiling (Epitahedron E±); Poincaré Homology Sphere;
_________________________________________________________________________________________

## 28 1. Introduction

The SciArt program *Resonances* of the Joint Research Center (JRC) in Ispra, as described by the editor Tiziana
Lanza in the introduction to this volume (Lanza et al. 2020), is guided by a noble motive, the establishment of a
convergence and harmonization of sciences and arts. So, the nowadays almost 70 year old campaign against the
'two cultures' (c.f. Snow, 1959) and pro integrating the resources of disciplines spanning the natural and social
sciences, the arts and the humanities, became finally alive in the current societal process triggered by SciArt initi-



atives of the European Commission. In a way, SciArt -- or information art, as it is named sometimes -- nowadays,
can be regarded as a restart of Romantic ideas (c.1800–1850) when foremost Diderot & d'Alembert in France,
German Idealist philosophers and Romantic poets in England dreamed of knowledge production in a union
between the arts and sciences. Georg W.F. Hegel (1770–1831) claimed that "in our age, art invites us to intellectu-
al consideration" (Hegel, [lectures, 1788]1975). Hegel spoke about opening up the possibility of a different kind
of art which is „explicitly self-reflexive and exploratory" (Žižek, 2016).
In this spirit, the British artist John Constable famously (1776–1837) demanded that painting be

"scientific as well as poetic … a branch of natural philosophy,
of which pictures are but experiments." (John Constable, 1836, in Beckett, 1970)

The mathematician Ada King, Countess of Lovelace (1815–52) famous for her conception of the first Computer
program (1842) praised the importance of imagination (Toole, 2010, 136). It is not the sole preserve of artists or
poets, but rather that true imagination only arises when conceptualizing the world using the principles of science
and mathematics (Illingworth, 2019).
Lovelace claimed a poetical science while Humphry Davy (1778–1829) and James C. Maxwell (1831–79) were
writing sonnets. Foundations for modern arts were laid by Maxwell's development of a three-color method for
photographs resulting from experiments conducted with his wife Katherine Mary Dewar (Maxwell, 1861b).
Contributions to the edifice of knowledge can be hardly allocated either to art or science exclusively: For ex-
ample, at the beginning of meteorology as modern science, the classifications of clouds goes back to the manufac-
turing chemist and amateur meteorologist Luke Howard (1772-1864) who painted clouds. He established the
terms Latin 'stratus'- 'cumulus' and 'cirrus' which are still in use today (Howard, 1803). It seems notable that
Howard's 1802 presentation took place at the Askesian Society, a largely *non-conformist* group dedicated to natur-
al and experimental philosophy. — Progress in science usually does not take place in midst of established convic-
tions and doctrines. Nevertheless the idea of a fusion between art & science remained a futurist project since
claimed by the German composer Richard Wagner (1813–1883) in his essay *The future of Art*, when he remarked
a strong  deficiency of acceptance,

"I've brought up the Philistine against me who just want to imagine the artist as silly, but never think-
ing." (Wagner, [1849] 2013).

Hence, it seems the separation of the two research cultures increased since the term 'scientist' was coined by then
most influential British polymath William Whewell (1794–1866). In his review of Mary Sommercville's best
selling book *On the Connexion of the Physical Sciences (1833)* he was referring to 'some ingenious gentlemen
who 'proposed that by analogy to *artist* they might form *scientist'* due to fact that physical science became end-
lessly subdivided (Whewell, 1834, 59).
Here we are also addressing the manifold intertwinings between arts and sciences: Art may contribute to scientific
research with imagination  and visual expertise in order to create complex models. Secondly, art as science com-
munication enables to include emotional aspects which are not allowed in rational objective science which may
help to understand the scientific facts better. A third but not easily accepted role of the arts are critical commen-
taries to science, like f.e. Marcel Duchamps'  works »The Large Glass« (1915–1923) or  »3 Standard Stoppages«
(1913-14) (see,  Henderson, 2005) . While data scientists are mostly calculating, and philosophers ask for causali-
ty and oncologic foundations, artists want to know how the system „looks like". The geometer investigates visual
mathematical shapes for the representation of phenomena.



**2.    GAIA 5.0: The Installation**

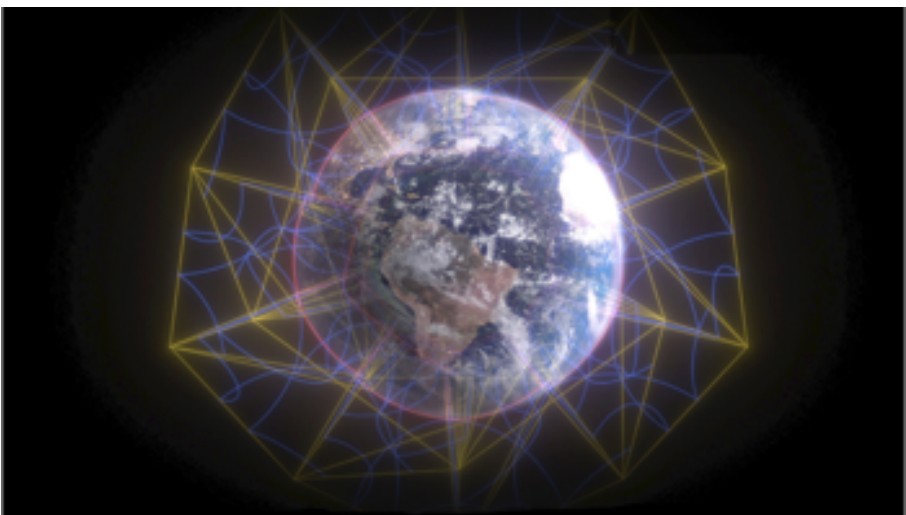


**Figure. 1**. The earth embedded in the hyper-Euclidean framework of 5-dimensional space. (Image: rendering
F. Grünberger for *GAIA 5.0* video by R. C-Z-Quehenberger,  still frame, min 2: 06 )

The SciArt installation, G*AIA 5.0*  a holographic projection and VR-experience of 3D animated film was commis-
sioned by the Joint Research Center (JRC) of the European Commission (EC) in Ispra and shown at the DATAMI
Resonances III festival with exhibitions in Ispra and at the Bozar in Brussels (Dec.2019-- Jan.2020) (datami,
2019). DATAMI is the portmanteau-word created by curator Freddy P. Grunert in order express his Resonances III
theme, addressing artistically societal challenge relating to big data and its reflection -- on  a traditional Japanese
tatami mat (Grunert--Fiordimela --Eeckels, 2020-21).
GAIA 5.0 was conceived after vivid discussions with two meteorologists. One was about Earth'  intrinsic
dynamics with Louise Arnal, an expert on floods,  affiliated at the  European Centre for Medium-range Weather
Forecasts (ECMWF) in Reading (UK) and the other was about the aerosol formation process with Frank Raes, a
meteorologist and former head of the Climate Change Unit at the JRC, the inventor of the SciArt program. Hence,
this artwork became a collaboration with meteorology, a branch of the atmospheric sciences which includes
atmospheric chemistry and atmospheric physics, with a major focus on weather forecasting.
**2.1 Motivation and target**
There is no general agreement on how to handle higher dimensions in mathematics which remain mainly ab-
stract. This indicates the necessity for contributions from the side of the arts and their expertise in imagination and
in generating images to overcome those shortcomings  in science.





The voice-over by Lydia Lunch in the video-art work creates the illusion that the Earth herself is narrating in the
spirit of the quote »Gaia is a tough bitch« (Margulis, 1995). Quehenberger's text reflects ancient and modern sci-
entific concepts spoken by the US-american singer, environmental activist and no-wave-icon Lydia Lunch who
also composed the music for it. Her powerful voice should echo the lament of the »World Soul« over the current
state of the world, known as »climate crisis«. The dramatic stage of the world, from severe cyclones to desertion,
from devastation to pollution  and the astronomically high rate of loss of bio-diversity in the Anthropocene is a
motivation for the production of art works which create devotion to the glory of nature wonder over the miracle of
life in the eyes of the spectator.  In this regard the artwork GAIA 5.0 promotes the beauty of symmetries in the
higher-dimensional space grid in which we and the stars around us are caught if we abandon the empty flat
/curved pseudo-4-dimensional Minkowski-spacetime paradigm.
In the title GAIA 5.0, the suffix 5.0 is referring to the 5th spacial dimension by mimicking the usual notion web
2.0  for interactive internet or web 3.0  for the semantic web and web 4.0 for the autonomous, content-generating
AI agents of the internet of things. *GAIA 5.0* refers to the Earth system functioning as a single living whole which
is more than a synthesis of any technological characteristics such as some utopian *web 5.0* currently under
development and referred to as the Sensory Web. In the DATAMI artwork the ancient geometry idea of a space
grid which is visualized by 5-dimensional geometry is morphing into the satellite grid. The voice-over explains,
"This imaginary sparkling space grid was replaced & actually realized
by means of technical devices forming the radiance grid of satellites
which now provides us continuously with surveillance data of earth' features:
both atmospheric and oceanic." (Lydia Lunch as Gaia narrator)
Thus in a way the piece is dedicated to a general audience as a meditation on different realities which emerged
over time: One of our contemporary high-info-tech existence with Earth sciences and observation technologies
and the other, millenia old mathematics, geometry, philosophy and imagination about the divine Earth,-- a living
creature embedded in the evidently higher-dimensional living pattern of the world.
It seems notable that the satellites used for telecommunication and observations are forming a technical 4-dimen-
sional environment, a metal shield around the planet. Global positioning systems (GPS) functions based on
Minkowski's pseudo-4-dimensional geometry but as a consequence no comprehensive 4-dimensional model of the
Earth exists.  But there are 4D weather data models with images of 'data towers' built on the surface of the globe
such as the Four-Dimensional Weather (4DWX) system which is well established since the 1990s (2018 RAL An-
nual Report).  In this regard the GAIA 5.0 video presents the Earth as hypersphere for visualizing space and time
as 4-dimensional object in (imaginary) movement (see fig. 1 and 2b).
Satellites are creating a technological web around the planet that replaced what was previously considered as
'living intelligent Pattern of the world' by Plato  (Platon, Timaios 29d–30c).
In 2019 the European Union's Earth environmental satellite *Copernicus* was launched. The *Copernicus* observa-
tion program provides important data for meteorology and weather forecast, the information obtained is data of
the Earth' atmosphere, seismic  and  oceanic features around the globe.  The depiction of satellites in the *GAIA 5.0*
video are purely symbolic. The number of actually launched space objects of nearly 10.000 is documented by
United Nations Office for Outer Space Affairs  (UNOOSA, Online Index, March 31, 2020).  Satellites which are
used for telecommunication and observations are forming a technical 4-dimensional environment, a metal shield





around the planet while no comprehensive 4-dimensional model of the Earth exists. Only a so-called 4D weather
data models (4DWX) create images that remind us of "data towers" built on the surface of the globe (c.f. 2018
RAL Annual Report).

## 2.2 The 4-dimensional hypersphere -- imagining the biosphere

The French philosopher Paul Virilio (1932--2018) attributed the negation of the notion of physical dimensions and
a lack of spacial awareness to a fractal perception of reality (Virilio, 1984). The discovery of fractals actually goes
back to the British physicist and meteorologist Louis Fry Richardson (1881–1953) who pioneered modern
numerical weather forecasting in the 1920s.

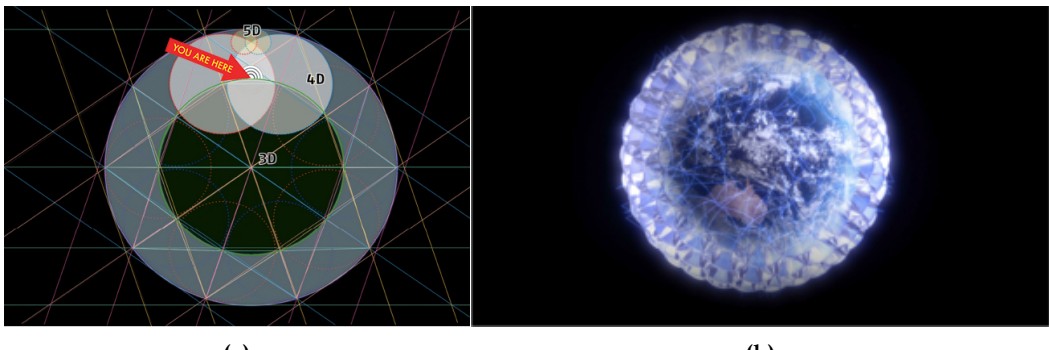

(a)                                                            (b)

**Figure. 2**. (a) The 5-dimensional space-grid and the embedding of the planet. image RCZQ; (b) GAIA 5.0,
the earth as hyper-sphere. Usually the hypersphere (S3) is not very well depicted and often declared as
„unimaginable". It is described as three-dimensional sphere S3 with the boundary of a disk in 4-dimensional
Euclidean space,-- there you find on each point of the 3D sphere the center of a small sphere with the imagi-
nary radius of the horizon. (Still frame, GAIA 5.0)

Fry Richardson first recognized fractal patterns in 1926 by studying the turbulence phenomenon at stream current
within wind wheels and coast lines (Richardson, 1926). More then 50 years later the notion „fractal" was coined
by the French-American mathematician Benoît Mandelbrot (1924–2010) who produced the first digitally
generated mathematical visualisation on an IBM computer, known as Mandelbrot set in1980. This lead to the
development of fractal geometry. Mandelbrot wrote a book on the fractal geometry of nature with examples of
self-similar fractal features like the branching of rivers, veins and neurons (Mandelbrot, 1977).
This lack of the human ontologic position as manifest in the absence of recognition of the 4-dimensional space
may well result in the lacking awareness for the biosphere we inhabiting on the surface of our sphere, currently
epitomized by the ecologic crisis. Fig. 2a depicts an inclusive position embedded into the higher-dimensional
realms where the 4D tangent spaces around the 3D sphere is the space of the biosphere.
The four-dimensional sphere is simply analogously to the four-dimensional hypercube constructed, which was





described as "a cube inside another" (Schlegel, 1882). The same mathematician, Victor Schlegel wrote a pamphlet
in which he forbids imagining the *4th dimension* to *artists* and *spiritists*  and claims to leave it to the
mathematicians as a purely abstract meaningless feature. (Schlegel, 1888) Alas,  his influence still resounds today:
There is no model for the 4-dimensional space and the hypersphere (S3) is declared as „unimaginable".
For the hyper-Euclidean model of the hypersphere as representation of the Earth's complex system, we take an
ordinary 3-dimensional sphere (in mathematics an ordinary 4D sphere is denoted as S2: a 2-sphere, because it
takes only the surface into account) and attach hemispheres on each point of its surface. So we get a complex 3-
dimensional representation of a sphere with a 4-dimensional 'real' 4D Euclidean 'tangent space' consisting of
interconnected, interchangeable hemispheres in imaginary movement that satisfy the Hamiltonian description.
**2.3 The video time-line**
The *GAIA 5.0* art-work mainly works on three layers of perception. One is purely esthetically attracting an
audience with no a priori knowledge of the subject. The other is communicating the scientific topics at heart (like
the $CO_2$ transport and tropical cyclone formation). A third motivation is to use the movie for proposing new
representations in meteorology and open fundamental research questions: how can we depict the geometry of the
Earth in a way that it includes the biosphere and what is the cause of Earth' geophysical and geological
dynamics ?
The timeline tells the story of Gaia embedded in a higher-dimensional space grid, narrating the original concept of
the ancient idea of the ether as conceptualized by Plato & Pythagorus, -- only recently visualized as infinite 5-
dimensional space  by Quehenberger's Quantum Cinema group (2010-13). Gaia displays the visualisation of the
large-scale patterns of carbon dioxide concentrations and transport of the year 2006 as shown in fig 3a and fig.6b.
(NASA, 2006). The subsequent transformation of the hypothetical space grid into the technological grid of
satellites leads us to the visualization of collected observational data of two severe cyclones that occurred in 2018.
The primordial planet depicted as Pangaea inside the dodecahedron refers to Johannes Kepler who assigned this
Platonic shape to  the trajectory of the planet Earth (Kepler, [1596] 2005, 16).  The film continues by explaining
how the chiral movement of the 5-dimensional space following Poincaré's description of the homology sphere
(Poincaré, 1904) leads to additional explanations for the formation of cyclones. This is usually ascribed to the
Coriolis effect (Coriolis, 1832) which evoked a conflict between common sense and mathematics for four
centuries (Persson, 2005). The text originally stated about the ether '… famously declared as superfluous in
1905.', but Lydia Lunch concluded intuitively, 'by Einstein  -- who got it wrong.'

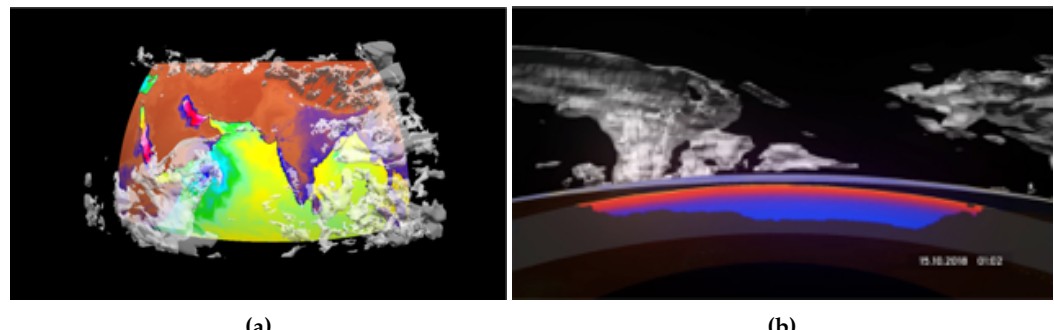

**(a)**                 **(b)**

**Figure. 3**. (a) Coupled METOC model, showing clouds & the temperature of the sea, colors range from 22°C(blue) to 34°C (red); Taiphoons over the Arabian Sea,  October 13, 2018, Image generated from ECMWF data with the COVISE/ OpenCOVER visualization program processed by Leyla Kern/HLRS  b) Taiphoons rendered ( Image: GAIA 5.0 video still frame, min 7:04) The   temperatures and the cross section of the ocean indicated temperature form red (24°C) to dark blue (10°C). from with clouds & humidity data; both images generated from ECMWF data with COVISE visualiza-tion program   earth embedded in the hyper-Euclidean framework of 5-dimensional space. (Image: rendering F. Grün-berger for *GAIA 5.0* video by R. C-Z-Quehenberger,  still frame, min 2: 06 )

## 2.4 The COVISE/OpenCOVER visualization of cyclones

The 2019 Resonances theme DATAMI, was dedicated to „big data". GAIA 5.0 uses coupled oceanic & atmospheric big data sets stored at the ECMWF of two very severe cyclones Luban & Titli that occurred in the Arabic sea and the Bay of Bengal October 2018.  The visualization makes the 16 days of the stored event in a 2 minute time lapse alive again. By displaying the dynamics of the atmosphere connected with the cooling of the oceanic water the question arises whether the Earth is deliberately producing cyclones as mechanism for cooling oceanic water systems.

The visualisation of the ECMWF reanalysis data of the two typhoons were realized through the open source software COVISE/OpenCOVER at the High-Performance Computing Center Stuttgart (HLRS) accomplished by Leyla Kern. The HLRS is famous for its big data processing capacities and visualisation expertise. Their software enables interactive exploration of immersive environments and supports the processing of large data sets.

The resulting rendering allows the visual combination of both, oceanic and atmospheric dynamics (Quehenberger, Arnal, Mogensen, 2020). It allows to visualize the stored data of this event with the following parameters: Longi-tude, latitude, geopotential height, surface pressure, ocean depth on specific areas & wind components. This si-multaneous visualisation seems important since ocean coupling in tropical cyclone forecasts will be even more important in the future, at higher atmospheric resolutions (Mogensen, 2017/18).

In the installation the coupled atmosphere-ocean visualization was executed as interactive holographic 3D big data visualisation model. Thus, the visitors of the exhibition can enjoy a virtual flight into the eyes of the cyclones by means of  a virtual reality equipment, such as VR glasses and  joysticks allowed to navigate through cloud formations.



### 3.     Scientific and philosophical background of the Gaia Hypothesis

### 3.1 The Gaia Hypothesis

The Gaia hypothesis states that Life is a planetary-scale phenomenon. The chemist James Lovelock (*1919) developed this idea during the 1960's when he was working for NASA searching Life on Mars. He further developed it with the microbiologist Lynn Margulis (1938–2011) suggesting the earth's atmosphere is "a component part of the biosphere rather than as a mere environment for life" (Lovelock & Margulis, 1974). Later Margulis brought autopoiesis (self-production), a clear-cut characterization of living systems (Maturana and Varela 1980) into Gaian science to the extent of presenting Gaia as an autopoietic system (Clarke, 2009).

> "The Gaia phenomenon is therefor, a *sui generis* biological system and the embodiment of Life itself in the planetary domain. [...] Hence, self-production by closure to efficient causation is more fundamental than self-regulation by feedback mechanisms." (Rubin & Crucifix, 2019)

This opposes the viewpoint of a necessity of a "Gaia 2.0" as claimed by the French philosopher Bruno Latour and the British earth system scientist Timothy Lenton, who claim,

> "Gaia has operated without foresight or planning on the part of organisms, but the evolution of humans and their technology are changing that" (Lenton and Latour, 2018).

Is this the voice of the overrated human consciousness (and therefore a presumed exceptionalism) (Sagan 2020), the hybris which lead to the Anthropocene -- the new planetary epoch in which humans have become the dominant force shaping Earth's bio-geophysical composition and processes ? -- We remain agnostic about the foresight of our planet, but the Life-Mind continuity thesis states that every biological phenomenon is a cognitive phenomenon (Maturana and Varela 1980, Friston, 2013). Hence, mind is prefigured in Life and belongs intrinsically to Life. Thus Gaia must be a cognitive system.

What Rubin & Crucifix call 'crucial' is that Life autopoietic organization is placed on a molecular level, and higher order autopoietic systems as Gaia should be produced through the autopoiesis of lower unities (cells and metacells) (Maturana, 1980). This would conclude a principle of Life as personified in Gaia, as uniting superior entity which in the last centuries and decades got replaced by Sky Gods and in cybernetics by the feedback omnipotent steersmen, lately.

The higher-dimensional geometry for the Earth is presenting an unifying view and thus contributing to current problems of representation in meteorology where scientists are calling for accurate maps of large regions of the sphere. The space model in GAIA 5.0 proposes 3-dimensional sub-spaces for the depiction of Gaia in contrast to current planar geoscience models which let them „keep shifting the tangent planes" as Rubin & Crucifix (2019) suggested it. For the first time Gaia is depicted as a hypersphere (fig. 3b), as the theoretical biologist Robert Rosen once mentioned:

> "The surface of the sphere is in some sense a limit of its planar approximations, but to specify it in this way requires a new concept (the topology of the sphere) that cannot be inferred from local planar maps alone." (Rosen, 1985)





### 3.2 Gaia in ancient Greek mythology and indigenous people


Mother Earth was allegedly ubiquitously worshipped during at least 40.000 years until during the troublesome
iron age when after all gods and goddess Astraea was fleeing from the new wickedness,  greed and violence of hu-
manity and  returned to the firmament according to Ovidius (Ovidius, [8 AD] 2008, 148f).  Nowadays indigenous
peoples f.e. of South America continue to call her Pachamama and still live in harmony with nature. They contin-
ue to have their non-mechanical agricultural activities.
*GAIA 5.0* is  also referring to ancient Greek mythology where the planet Earth was represented by the primordial
Goddess Gaia, Mother Earth who reigned with her son and husband Uranus, the God of the Sky.—- Some say he
was conceived by Gaia, who would become his wife, while others say that he was the son of Aether and Gaia who
as herself born from love and light. Detroned by their children, a patriarchal dominated society established the
Dodekatheoi twelve Goddesses and Gods residing on Mount Olympus, during the 6th century b.c, . The number
12 seems again to refers to the twelve sides of the dodecahedron and the seat of the Gods as 'thin upper region of
air' synonymous with the ether.  The ether was named Greek 'fifth element' by the Greek philosopher Plato
(428-348 B.C.) and "quintessence" by his scholar Aristotle. The search for this mysterious substance became a
century long target of Alchemists and moved into modern science as *luminiferous ether* but remained „undetect-
ed", before it was eliminated from science books in the 20th century.
The  'fifth element', known as fifth regular solid, the dodecahedron  was only described in a cryptic note, by  Plato
'yet there is a 5th combination [of triangles] which  the god used in the delineation of the Universe.'
(Plato/ Timaios, 52b)
Only recently Plato's riddle of the 5th element could be solved by identifying the shape of the dodecahedron as
compositions of unit cells of 5-dimensional space (Quehenberger, 2016). This gives rise to the assumption that the
ancient 'geometry of the Earth' was originally conceptualized as hyper-Euclidean. The word geometry stems from
the Ancient Greek: γεωμετρία; geo-'earth", -metron „measurement' and indicates geometry as a way for the
perception of the world.  It is a branch of mathematics concerned with questions of shape, size, relative position
of figures, and the properties of space.
In Greek mythology Mnēmosýnē, (from mnēmē; Greek for memory) the mother of the Muses is the one who
reminds her mother, Gaia, the earth. Geometry and grammar are attributed to the Muse Polymnia, also protector
of the divine hymns and the imitative arts. Thus, etymology unites the divine feminine, the Earth, geometry and
the arts and the ancient idea of the ether, recently identified as 5-dimensional space concept (Quehenberger,
2019). In texts by Aeschylus (c. 525/524 – c. 456/455 b.c)  and Pindar (c.518-- 446 b.c.) the ether was also treated
meteorologically, namely as that layer 'in the etheric circle, a seat where the foggy cloud turns to snow' (Fürlinger,
1948). -- This means the ancients assumed space itself as responsible for aerosol formation just as visualized here
by means of the 5-dimensional space.

### 3.2  Geometrical figures used in *GAIA 5.0*


The currently used numerical weather simulations are based on temperature differences and thermodynamics, c.f.
how water or air moves as it is either heated by absorbing sunlight or cooled by emitting infrared  radiation, as





well as how such processes operate over a rotating, spherical surface. All dynamical atmospheric effects are based
on the assumption that heating and cooling are heterogeneous in space and time because land and sea have
distinct heat capacities and dynamics and the process of forecasting starts with the initial conditions based on the
best possible description of the current state of the atmosphere, land surface and oceans (Bauer, 2016). In addition
to this the GAIA 5.0 video proposes the artistic assumption of an underlying spacial geometry which could be
responsible for the tectonic and meteorologic dynamics on earth.

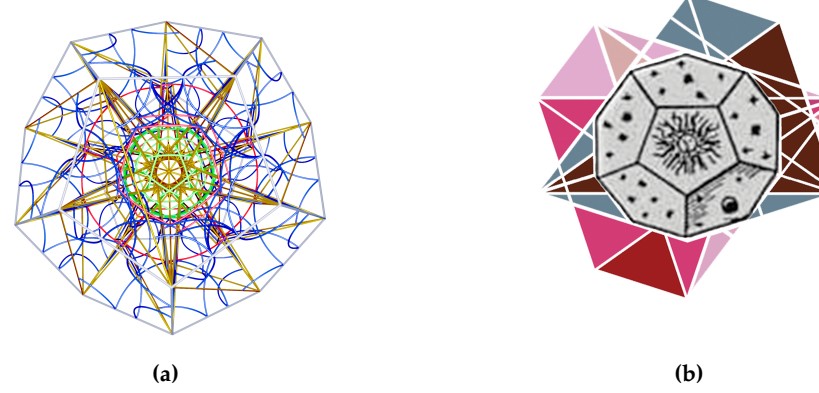

**(a)**                                     **(b)**


**Figure. 4**. (a) The *GAIA 5.0* web — Epta-dodecahedron, a 4-dimensional dodecahedron in a 72° twist  (rczq,
2013) (b) Two epitahedra forming the 3-dimensional dodecahedron, as originally described as  "5th element"
by Plato (rediscovered by Quehenberger), symbol for the Universe (dodecahedron in the center, drawing by
J. Kepler) drawing: R. Quehenberger)
The newly discovered 5-dimensional space cells are the 3D representation of the Penrose kites & darts tiling
(Gardner, 1977), the irregular pattern of the golden ratio ($1/2(1\pm \sqrt{5})$ = Phi = 1,681… or phi = 0, 618…) and
related logarithmic spirals have scientific significance (f.e. in astronomy, biology, genetics, etc.) featured by many
popular 'golden ratio' books. (in the plane, named Epitahedra (E±): Two epitahedra are forming the 3-dimensional
dodecahedron usually mentioned in schoolbooks as Platonic solid (fig.4b) and twelve epitahedra confine the 4-
dimensional dodecahedron, named epita-dodecahedron (Quehenberger, 2013). All 4-dimensional Platonic solids
were constructed before by the British mathematician Alicia Stott Boole (1900).
The epita-dodecahedron works like the 4-dimensional dodecahedral space as conceptualized by the French
polymath and last universalist, the philosopher, mathematician, engineer and physicist Henri Poincaré (1854–
1912) who suggested it as model for the Universe, here probed as model for the Earth (Poincaré 1904). It is the
infinite 5-dimensional space that works like a "symmetry machine" for the generation of particles. Here it is
applied to the planet Earth and aims to explain the chiral dynamics which occur in the formation of cyclones as
well for the formation of aerosol particles. This 3-dimensional formation of the golden ratio seems logic, insofar
as the logarithmic spiral is a generation method for Penrose Patterns, the golden tiling of the plane (Kappraff,
2001).  These patterns  look the same at every scale (Penrose, 1974). Ultimate chaos results in the pattern of the
Golden ratio. Hence both characteristics of cyclones, their fractal and chiral nature could be explained by the
assumption of this underlying dynamic higher-dimensional space.



### 3.2 Meteorological aspects of the art work

The chiral motion of the Penrose tiles and the chiral counter-movements of the 3D tiles wich are confining the
epita-dodecahedron is depicted in the geometry of GAIA 5.0. In fact it would present many solutions to forgotten
problems, like the chiral movement of galaxies and cyclones, and seeds of sunflowers. Similarly we find in the
turbulence of air in the vortices of taiphoons in smaller regions of air. The empirical logarithmic approximation is
shown with several examples of satellite and radar images of a tropical cyclone (TC) (Yurchak, 2007).
In meteorology the chiral movement of the water and air is described as heat transport to the atmosphere, vertical
mixing in the ocean, and upwelling by Ekman pumping (Ekman, 1905). Walfrid Ekman (1874–1954) laid the
mathematics for Bjerknes's theory of the current moving along an inward, clockwise spiral (Ekman 1905). Gaia
5.0 proposes that the Ekman spiral could lie inside the hemisphere in a fraction of the epita-dodecahedral space.
Could the dynamics of space itself interfere with hydrodynamics and geo-physical motion? The same effect is
responsible for cyclones. -- So the question is: Do these dynamics of space itself interfere with hydrodynamics
and geo-physical motion?

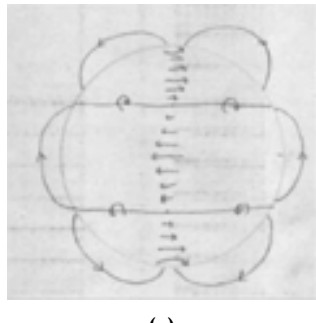

**(a)**          **(b)**
**Figure. 5**. (a) V. Bjerknes „Vortex tubes in the atmosphere", idealized sketch of the three-dimensional global
circulation (in pencil) drawing from lecture notes of 1917, Section 82 (b) Side-view of the epita-dodecahe-
dron with counter-rotations of the pentagonal faces according to descriptions for the Poincaré homology
sphere, (upper side to the left, lower side to the right ) image: RCZQ, 2012.
In meteorology similar considerations were pondered around 1900. Incidentally the dynamic epita-dodecahedron
model corresponds to an early ether model (fig. 6.a) conceived by one of the founders of modern meteorology and
oceanography, Ekman's teacher, Vilhelm Bjerknes (1862–1951). He adapted the hydrodynamic theory of the ether
(V. Bjerknes, 1900) by his father Carl Anton Bjerknes (1825 --1903) as an application to meteorology (V. Bjerk-
nes, 1898). As a matter of fact some basic principles of modern meteorology are derived from his model which
comprises „vortex tubes in the atmosphere" which reminds us of the vorticity lying in the counter-movement of
the epitahedra inside the 4-dimensional dodecahedral structure.
Generally speaking, British scientists of the 19th century who laid the foundations of meteorology favored the





existence of an all-pervasive ether. Also many artists were fond of the ether in the early twentieth century. The
values of modernism was found in the complexities and contradictions of modern physics provided a fertile
ground for the development of new artistic languages (Navarro, 2018). Once more, the here described artwork
refers to it in the  shape of the 5-dimensional space as follow up principle proposing new / old ontological
foundations.
**3.3 Pangaea  -- Kepler**
Meteorology uses icosahedral models in order to stretch out the surface of the globe onto a plane as flat triangular
pieces.  Tthe world map projected onto the surface of a three-dimensional icosahedron that can be unfolded and
flattened to two dimensions was first presented as *Dymaxion Map* by the US-american geometer and architect R.
Buckminster Fuller (1895–1983)  in 1943. Almost 20 years later in meteorology the icosahedral grid was
proposed originally during the 1960s by Gates and Riegel (1962 & 1963) a.o. It was chosen as model for
subdividing a sphere because it has the highest degree of symmetry for the design of an equal-area grid.
The dodecahedron is the geometrical dual of the icosahedron. The dodecahedron as shape for the Earth is
mentioned by Plato when he compares the planet with a piece of art that resembles a colorful football (Plato,
Phaidôn, 110b). Also the mathematician  Johannes Kepler (1571-1630) who laid the foundations for modern
astronomy assigned the dodecahedron to the Earth (Kepler, 1596).  This gave rise to a beautiful picture of the
primordial earth,  Pangaea embedded in the small dodecahedron in the center of the epita-dodecahedron, fig.6a.

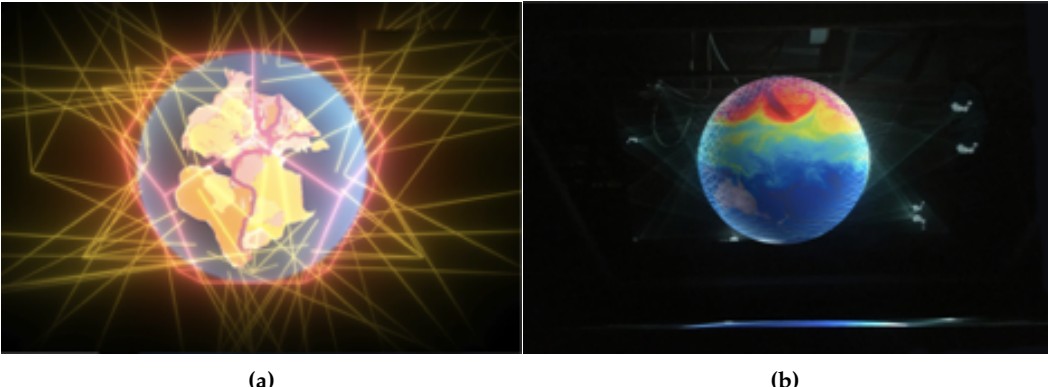

(a)                                                     (b)

**Figure. 6**.  (a) The planet Earth inside the small dodecahedron in the center of the 4-dimensional dodecahe-
dron, with the face of the supercontinent. Pangaea, (Image: GAIA 5.0 video still fame, min 1:33; animation
by F. Grünberger, 2019); (b) CO2 transport simulation projected of the globe with surrounded by a bunch of
satellites (Image: GAIA 5.0 video still fame, min 3:58, animation, by F. Grünberger; holographic projection
at the DATAMI - Resonances III, Festival at the JRC in Ispra)
In order to express the living system most lively we start with image of the primordial landmass, named Pangaea
(Wegener, 1912). Pangaea's outline reminds us of the shape of a female figure, who holds a baby in her left arm



while her head in a sleeping position is directed to the North Pole.
This film sequence is shaded with Lydia Lunch's velvet voice reciting:

"Pangaea, a dreaming living creature, dreaming of living creatures."

The video animation shows planet Earth  in a time-lapse evolution from the primordial state as supercontinent
Pangaea, meaning 'All Earth', began to break apart about 200 million years ago, until the present positions of con-
tinents is reached. Plate tectonics theory predicts the earth`s land mass will again form another supercontinent
in the future when the Americas will be joining Asia from the eastern side in the middle of the Pacific ocean,
in another 230 million years. Compared to a human life cycle this would mean the Earth is now— a 45 year old—
in the middle of her life span and yet climate change and the extinction of species is endangering her complete-
ness & beauty depending on the diversity of life. The *Panagea*- image in the video is taken from Alfred Wegener's
(1880–1930) vision of *Pangaea*, the German polar researcher, geophysicist and meteorologist who coined the
name for the primordial earth and first hypothesized the continental drift theory (Wegener,1912). The video ani-
mation shows a slow displacement of the continents from the *Panagea* state to the present position: Afrika made a
36° counter-clock-wise turn, while South America has turned 36° clock-wise, the Pacific side of North America &
Alaska  is the 72° counter-clock-wise flip of the *Pangaea* picture 72°. The same rotation angels are inherent in the
5-dimensional dodecahedral space and seem to undermine the assumption of a subscendent space dynamics.
GAIA 5.0 shows a poetic image of the Earth embedded in the center of this infinite 5-dimensional space inside the
highlighted shape of the dodecahedron. -- Actually this is a simple illustration and not the accurate picture,
because Kepler who also considered the Earth as living entity speaks of the relations of the trajectory of the planet
lying on a dodecahedron. Meanwhile the symmetries of the Kepler problem was solved by showing that the
equation for the conservation of energy can be written the same like of a sphere in 4-dimensional space
(Göransson, 2015).  Then the ellipses would be circles on a 4-dimensional sphere and only appear elliptical on a
projection on the plane (Baez, 2015).--  We may also assume that Kepler was studying the relation of polyhedra in
the center of the 5-dimensional space as first visualized during the Quantum Cinema project (Quehenberger,
2013).  He was asking for the cause of movement of planets, the 6-fold structure of honeycombs and hexagonal
snow flakes and used the same geometrical patterns and solids (Kepler, 1611). The fact that his seminal discovery
of the three celestial laws is based on artistic practices are still completely ignored by contemporary astronomy.
His astronomical findings are not only based on 2D pattern designs and regular solids and investigations on space-
filling shapes but also on musical relations. In order to find the appropriate shape of the space which enables all
sorts of matter, he already operated with hyper-Euclidean geometry and designed 4-dimensional polyhedra which
were re-discovered twice as models for quasicrystals living in 5- or 10-dimensional spaces (c.f. Levine &
Steinhardt, 1986). This means that not Newton but Descartes (Newton, 1666) and Kepler could support the
geometry underlying the planet Earth as featured in the GAIA 5.0 movie. Renée Descartes (1596–1650)
mentioned first interconnected coordinate frames for the representation of higher dimensional spaces  (Descartes,
[1637], 1902). He proposed that all matter is just a result of transformations of spaces is Pythagorean and
precedes modern Group Theory for 200 years. On the contrary Isaac Newton thought in terms of point mechanics,
speed and direction  (Newton, 1699).
What is remarkable about the higher-dimensional space configurations is that they must be regarded as 3D spaces





in movement. Space itself becomes a perpetuum mobile in dim < 3.  -- This new mode which is relying on
Poincaré's and Brower's concepts about higher dimensional spaces naturally opposes Minkowski's pseudo 4-
dimensional concept which is currently successfully used for GPS systems but also for natural phenomena.
The English philosopher Margaret Cavendish, Duchess of Newcastle (1623-1673) was the first who argued that
this gender constructed science hinders all that will lead to an objective epistemology. By redefining gender in
Cavendish's theory of matter she claims that there is no rest in Nature, and that this constant movement is not
induced by an external force because nature is an active, moving, powerful being, capable of movement within
itself (Walters, 2014 ).
**3.4 Chaos Theory,  the Lorenz attractor & non-linearity in atmospheric fluctuations**
Henri Poincaré not only first predicted gravity waves (Poincaré, 1905) but he was also the first proponent of
chaos theory in the 1880s (Poincaré, 1890). Poincaré even proposed that that such phenomena could be common,
for example, in meteorology. This happened 70 years later when the MIT-meteorologist Edward Lorenz
developed the first chaotic attractor from it (Lorenz, 1963). The Lorenz attractor has been almost ignored for a
decade but got a household name after he coined the term „butterfly effect" in order to illustrate the sensitive
dependence on initial conditions:
'If a single flap of a butterfly's wing can be instrumental in generating a tornado, so also can all the
previous and subsequent flaps of its wings…….More generally, I am proposing that over the years
minuscule disturbances neither increase nor decrease the frequency of occurrences of various weather
events such as tornados.' (Lorenz, 1972)

The Lorenz attractor is a strange attractor, a fractal, and a self-excited attractor with respect to all three equilibria
is a set of chaotic solutions of the Lorenz system. The atmosphere is a highly dynamical system, and exhibits
many chaotic features. An operational weather forecasting model tries to model an initial value problem, in fact
one of the most famous examples of a chaotic system. His theory and application of chaotic dynamics was derived
from a simplified model of convection in the earth's atmosphere.  It also arises naturally in dynamos, models of
lasers and electric circuits (Haken,1975).  This  effect was also considered for the stable circular limit cycle which
appears in the twisting perturbation of electro-magnetic forces in the space of 5-dimensions, known as light in its
dual form as wave and particle (Quehenberger, 2012).
Chaos theory tells us that the critical value of both equilibrium points lose stability through a subcritical Hopf
bifurcation (Hirsch, Smale, Devaney, 2003). Hopf bifurcation (also sometimes called Poincaré-Andronov-Hopf
bifurcation) is a critical point where a system's stability switches and a periodic solution arises.
Recently the JRC researcher Daniel Tirelli made an analogy between a mechanical system (cable) and thermo-
hydraulic system (atmosphere) where you have an increasing input of energy: for cables, the multiple solutions in
chaotic systems, is represented by the existence of multiple basins of energy, which are the number of modes of
vibrations activated during its motion (Peng, et al.). As they are not unique, they can absorb more energy by
multiplicity than in classical mechanical system (one solution only).  Moreover, for the same displacement of the
cable as these new modes have higher frequencies they can absorb, each one, more energy due to the higher speed
of the cable (Tirelli, 2013). Therefore, the analogy is nearly perfect and could be resume in a multiplication of




small and important atmospheric events, (storms) for the atmosphere, and more modes activated for the cables.
Then for each event, more extreme phenomena (wind velocity, hail dimension etc. greater) for the atmosphere,
and faster motion for the cable (modes of higher frequency). This analogy is due to the presence of non-linearity
in both systems, cables and atmosphere flows.
In the time of the dramatic vision of the atmospheric changes, this window of 'non linearity' of phenomena gives
however a rather optimistic view of the global change. In effect comparing again the chaotic behavior of human
heart motion to the chaotic behavior of the climate we can observe the following: in most of cardiac pathologies
(restriction of flows) the heart need to push the blood with a higher speed to oxygen each part of the body. Its
results in a greater contraction of the muscle, which can reach the break. The sudden changes of cardiac rhythm,
commanded by the brain is a safety reaction to avoid this type of cardiac crisis. The behavior changes from cyclic
to chaotic for safety reason! -- Seeing more analogies between the human body and atmospheric behavior will
help us to perceive the Earth as living system.
## 4.  The dynamic formation of aerosol particles
Finally, with a zoom into atmospheric micro-processes the GAIA 5.0 video proposes a geometrical aerosol forma-
tion model based on the hypothesis that the same underlying dynamic geometrical framework is responsible for
the formation process of aerosol particles on the minutest level. The scientific idea of 'a higher multidimensional
view' was presented  in a  "3-D global Chemical Transport Model and in a General Circulation Model" by JRC -
scientists seemed to coincide well with the here described intention for the visualization (Raes, van Dingenen, et
al. 2000).

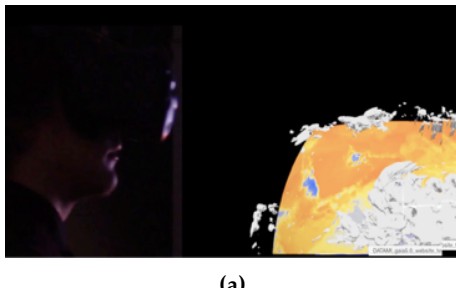
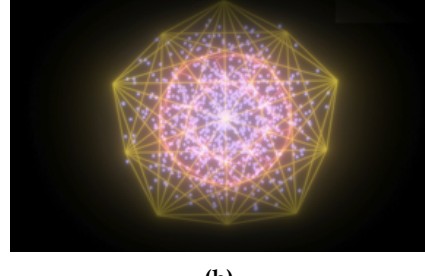

**(a)** **(b)**

**Figure. 7**. (a) Visitor with VR glasses navigating through clouds of the COVSE visualisation, at Bozar, Brus-
sels Jan. 2020 (image RCZQ)  (b) he icosahedron model wo epitahedra forming an epitahedron in the center,
here suggested as aerosol formation model (*GAIA 5.0* video. 8:37min)
The higher-dimensional aerosol formation model is inspired by the visualization of  the most simple Galois group
by permutations of one epitahedron in the framework of an icosahedron (Kostant, 2005). The permutation of one
epitahedron performing rotations over a Hamiltonian path over 11 vertices of the icosahedron creates visually a
hyperbolic dodecahedron (see red outline in fig.7b) which suggests that a 'new inner space' is created. So we can





imagine that the discrete space works like mathematical machine that not only produces symmetries as appropri-
ate for group theory to describe the formation of subatomic particles but works also as a 'mixer' which forms
aerosol particles dynamically.
This dynamic epitahedron model corresponds also to currently used fullerene (C60 and C70) model compounds
for the simulation of molecular dynamics which lead to nanometer-sized particle formation and growth in the at-
mosphere. (c.f. Liu et al., 2016). Due to a symmetry convergence the C60 structure can be replaced by a dynamic
epitahedron model in the framework of an icosahedron. Our proposed model has the advantage of inherent dy-
namics which allow a dynamical simulation and works on the assumption of a higher dimensional spacial struc-
ture as aerosol particle formation generator.
**5. Discussion and conclusion**
The exhibition in Ispra at the JRC with 2000 estimated visitors attracted mainly the experts of the different re-
search groups at the laboratories of the JRC. Whereas the Resonances III exhibition at the BozarLab in the center
of Brussels was counting 1100 dedicated visitors besides more than 20k visitors passing single exhibit at central
location in famous Bozar Centre for Fine Arts. In March the GAIA 5.0 art work was presented at the Milan Dig-
ital Week 2020 as online event with 1000 viewers.
Around 300 persons attended the opening event. The visitors of the GAIA 5.0 had much fun navigating through
the clouds and cyclones enabled by the virtual reality (VR) equipment while others could see the full video as
holographic projection. Discussions on higher dimensions and their representation as well as on weather models
based on Coriolis force, differential heating and cooling took place.
Certainly many discussions on higher dimensions took place with interested exhibition visitors who demanded
clarifications: Eg.: Gustav T. Fechner's 'shadow-beeings' inspired Edwin Abbott Abbott (1838--1926) to his
influential mathematical satire «Flatland -- A Romance of Many Dimensions« (1884) in order to illustrate that
mathematicians are hardly able to imagine a sphere, not to speak of any higher dimensional object. Ironically
»Flatland« is still referred to as 'proof' of a 'natural in-imaginability of higher dimensions' in mathematics books,
-- despite Abbott's plea,
"Yet I exist in the hope that these memoirs ... may find their way to the minds of humanity in Some
Dimension, and may stir up a race of rebels who shall refuse to be confined to limited Dimensionali-
ty." (Abbott, 1884)
Similary art influenced science when another novel, namely H. G. Wells' »Time Machine«, with the fiction of
travel on the 'time'-axis towards a distant future (Wells, 1895). Also the expanding universe was inspired by
Edgar Allen Poe's »Eureka« (Poe, 1848).
The discussions with visitors also addressed the feminist perspective on current scientific and technological
developments as a result of a 400 years long focus on empirical studies which successfully cut out all non-
observable entities as well as women from science. This was already discussed in the 16th century between
Francis Bacon (1561–1626) who established materialistic philosophy (Bacon, 1620) and the feminist anti-
empiricist, Margaret Cavendish. She ascribed the pejorative approach to nature to the Aristotelian description of
the female applied to the conception of nature as merely passive, empty, lifeless female body (Cavendish, 1664).





In contrast to the prevailing materialistic approach in science the philosopher Giordano Bruno (1548–1600) — who got burned by the Catholic church a.o. for his concept of infinite and many worlds, — suggested to regard nature herself as artist (Bruno, [1584] 2015). Bruno cited explicitly female methods of production and also addressed the misogynous reservations of his time responsible for the maltreatment of nature — those who think badly of women cannot be able to worship nature in her female role. Shouldn't she who designs and creates everything considered to be the greatest artist?

— In summary, the audience understood that art can be a tool for creating a visual and audible, a sensuous access to the perception of meteorologic dynamics and the complex space we inhabit. The perennial concept of autonomously dynamical higher-dimensional spaces, Plato's theory of everything renders valid in the light of 5-dimensional hyper-Euclidean geometry. This shall help to understand not only the formation of aerosols and cyclones, but also the perpetual motion of the Earth and the entire Universe, for which there is currently no other valid explanation. If GAIA 5.0 can produce in the minds of the spectators some humble amazement for the wonders of the animate world in order to handle it with care, the purpose of this art-work was achieved.

**Supplementary Materials:** C.-Z.-Quehenberger, R.: GAIA 5.0 — A Holographic Image - Ambience, film: https://doi.org/10.5446/49792

**Acknowledgments:** The philosopher, here artist, R. C.-Z.-Quehenberger expresses her gratitude to all collaborators in the art world, the singer Lydia Lunch and the VX artist Florian Grünberger as well in the science world, meteorologists Rita Van Dingenen, Thomas Petroliagkis, Jutta Thielen-del-Pozo and Frank Raes at the JRC and Louise Arnal at the European Centre for Medium-range Weather Forecasts (ECMWF), Reading, UK; and SciArt experts Leyla Kern and Dr.-Ing. Uwe Wössner for COVISE/OpenCOVER visualizations at the High Performance Computing Center (HRLS) in Stuttgart communicated by A.Wierse (SOCOS) during this DATAMI-Resonances III SciArt research project. Parts of the here presented hyper-Euclidean representation of higher mathematics was previously developed by her for the visualization of quantum phenomena during the SciArt project *Quantum Cinema - a digital vision* (2010-2013).

**Author contributions**: S.R. contributed statements concerning Gaia hypothesis and the general presentation of the article, D.T. contributed his observations on cables and chaos theory applied to climate science and L.K. visualized the ECMWF data of the cyclones;



**539**     **Appendix A**

**540**     Text of the video GAIA 5.0 narrated by Lydia Lunch:

**541**

**542**     There is this boundary space between zero and one:

**543**     Series of roots of ones fractioned to infinity: phi --

**544**
$$\Phi = \sqrt{1+\sqrt{1+\sqrt{1+\sqrt{1+\ldots}}}}$$

**545**     famously a topic of many obscure speculations on the foundations of the Universe.

**546**     In the context of the infinite 5-dimensional space

**547**     which works like a machine that cuts each piece into Δ golden triangles

**548**     it may well gain a new/ old significance.

**549**     Plato's ascribed the permutations of these triangles to the wet nurse of becoming

**550**     who cradles them into matter  in all sorts of symmetries, colors, states of aggregation

**551**      & frequencies, forming the living intelligent pattern of the world.

**552**     In the center of this infinite 5D space machine

**553**     we find the geometrical shape of the dodecahedron,

**554**     a regular 12-sided shape that the astronomer Johannes Kepler assigned to the earth

**555**     & before him Plato & Pythagorus

**556**     Pangaea, the dreaming creature, dreaming of living creatures living in  living creatures

**557**     Plato named the construction of triangles which is forming this shape "the 5th

**558**     element", Aristotle spoke of it as „quintessence"—also known as the all permeating luminiferous

**559**     aether —synonymous for the seat of the gods & electromagnetic forces,

**560**     the light  & intelligence; ---

**561**     famously declared as superfluous in 1905 by Einstein  -- who was wrong.

**562**     The visualization of the hypersphere S3 by means of the unit cells

**563**     of 5-dimensional space  conforms to the Poincaré homology sphere

**564**      which gives rise to new interpretation of the earth' intrinsic dynamic

**565**     relying  on the geometry of space itself.

**566**     — Where would the dynamics stem from, other-wise?

**567**     In meteorology the Coriolis force causes moving objects on the surface

**568**     of the Earth to be deflected to the right in the Northern Hemisphere

**569**     and to the left in the Southern Hemisphere.

**570**     This corresponds to Poincarè's description of counter-movements of the pentagons

**571**     of the dodecahedron, here visualized by epitahedra.

**572**     The same effect is responsible for the rotation of large cyclones.

**573**     This imaginary sparkling space grid was replaced & actually  realized

**574**     by means of technical devices forming the radiance grid of satellites

**575**     which now provides us continuously with surveillance data of earth' features:

**576**     both atmospheric and oceanic.

**577**     Two very severe cyclones  Titli & Luban  formed simultaneously

**578**     in the central Arabian Sea.





Moving upon ashore, Cyclone Luban produced flooding rains
in Somalia, Oman, and Yemen.
The storm by Wednesday October 11 had reached wind speeds of 119-137km/hr
gusting up to 160km/hr
This was all over the  west central Arabian Sea around the system's centre.
The direct impact of these storms coming together
with heavy rains and strong winds
affected the coastline of Oman on the Arabian Peninsula starting from October 13, 2018
until they actually struck Yemen on October 14 in the midst of
not only a civil war but a cholera outbreak.
The storm killed 24 people in the country and injured another 124 people.
When Titli made landfall  in the south-Indian state of Andhra Pradesh,
at peak intensity and affected more then 5.7 million people &
killed at least 85 due to heavy flooding and landslides
causing damage of US$ 920 million in South and West Bengal in total.
The life period of the system was 210 hours
very unusual a rare occurrence post-monoon season.
Based on an underlying 5-dimensional space grid
we may also model a dynamic genesis of aerosol particles:
We take the icosahedral symmetries and use what Plato called „5th element"
higher dimensional spatial structure as formation generator.
It is inspired by fullerene (C60 and C70) model compounds
for the simulation of molecular dynamics.
These spacial features may well subscend dynamics in the air
that shall lead to nanometer-sized particle formation and growth in the atmosphere.
Thus we are in the midst of a living pattern
that forms life & light & dust in the same way.

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
