# Peer review of "GAIA 5.0 — A five-dimensional geometry for the 3D visualization of Earth' climate complexity"

_Geoscience Communication, 2020_

## Referee Comment (RC1) · Bruce Clarke (Referee) · 27 Nov 2020

Reviewer: Bruce Clarke Paul Whitfield Horn Professor of Literature and Science 2019 NASA/Blumberg Library of Congress Chair in Astrobiology

GAIA 5.0 — A five-dimensional geometry for the 3D visualization of Earth' climate complexity

General comments

Taken together, "Gaia 5.0" the artwork and the scholarly article combine to form a massively synthetic project for the re-convergence of art and philosophy with science and

mathematics. "Gaia 5.0" marshals the historical discourse of higher-dimensional math and physics to the visualization of a living planet captured by the cosmic vibrations of primeval geometrical patterns. This is inspired work built upon deep developments in Western thought. It is argued that Platonic-Pythagorean hyper-geometries bring out a kind of living depth appreciated by Kepler but missed by the Aristotelian tradition as it passes from Newton to Einstein. This shift restores the luminiferous aether discarded by Einstein by reviving its relevance for hyperspace descriptions. "Gaia 5.0" directs this research not at an abstract mapping of 4-dimensional space-time but directly toward a 5-dimensional moving image filling out the dynamical substance of Gaia itself—Earth's biosphere caught up in the formative play of geometric and physical forces.

Specific comments

I would have liked to see some commentary provided on the following provocative and scientifically promising statement: "By displaying the dynamics of the atmosphere connected with the cooling of the oceanic water the question arises whether the Earth is deliberately producing cyclones as mechanism for cooling oceanic water systems" (ll.201-3). This suggestion about Earth in a period of global heating using an increase in hurricanes to cool itself—an approach underscored by Lovelock's longtime insistence that "Gaia likes it cool"–is precisely what was implied when the authors characterized Margulis's autopoietic Gaia as a "cognitive system." Is there a way to connect Gaia's planetary cognition more tightly to the 5-D framework of the discussion?

Technical corrections

The vigor of the presentation is commendable, but the quality of the English in this article needs improvement at the level of grammatical correctness.

l.72: "oncologic" should be "ontologic"

l.386: "angels" should be "angles"

Sagan 2020 is missing from the references.

Please also note the supplement to this comment:
https://gc.copernicus.org/preprints/gc-2020-27/gc-2020-27-RC1-supplement.pdf

---

## Referee Comment (RC2) · Bruce Clarke (Referee) · 1 Dec 2020

Dear Renate Quehenberger: It is my pleasure to discuss this article with you. With regard to my response, I would also recommend your soliciting the advice of your co-author and Gaia theory specialist, Sergio Rubin.

I want to make sure I follow your sense here: you write, "Thank you for encouraging us to strike out the aspect of Gaia as a 'cognitive system.'" In English, the verb phrase "strike out" typically means, "eliminate." But I take you to mean, "emphasize" or "under-line": "Thank you for encouraging us to emphasize the aspect of Gaia as a 'cognitive system.'" I certainly do not mean that you should eliminate this aspect, but rather, that

you should take it farther.

Now, I need to clarify that my use of the term "cognitive" in no way means or implies "conscious." I would not think of asking you to grapple with the "origin of consciousness"! Rather, my usage stems from autopoietic theory (Maturana and Varela, as adapted by Margulis), in which context "cognitive" means "sentient," "responsive to its environment," or more poetically, "able to bring forth a world." In this sense, a living cell - say, a bacterium - is cognitive, and so, too, is "autopoietic Gaia."

For "autopoietic Gaia," then, such (arguably) responsive or compensatory (self-regulatory) behavior as generating hurricanes to diminish heat gradients is no more "deliberate" than any other manifestation of the second law of thermodynamics in a complex dynamical system. Nonetheless, the effect is to produce what the system "wants" - to cool itself!

So, what I mean by "planetary cognition" is just this registration by the Gaian system of its forcing by solar and terrestrial events by producing an "anticipatory" response. I dilate on this matter in my blog post "Cognition Not Consciousness": https://www.gaian.systems/research/cognition-not-consciousness. My question for you, then, is whether you might also see a way to invoke the hyper-geometrical "formal gradients" that your art-science project explores in support of this autopoietic model of planetary sentience, or sensitivity.

---

## Author Comment (AC1) · 1 Dec 2020

"GAIA 5.0 – A five-dimensional geometry for the 3D visualization of Earth' climate complexity"

Reply to referee Bruce Clarke's Interactive comment, by Renate Quehenberger 01, December 2020

Dear Prof. Clarke,

I am delighted that you've accepted to review our article and moreover about your overall positive feedback.

[Figure]

Thank you very much your suggestions concerning misspellings & for even checking the references.

I find especially your specific comments extremely valuable: ad 1) I am glad you found that the assumption the Earth could "deliberately" produce cyclones as mechanism for cooling oceanic water systems could be a "scientifically promising statement".

Thank you for encouraging us to strike out the aspect of Gaia as a "cognitive system" ! I am glad that I may quote your review and will add a paragraph with your references to Margulis and Lovelock's "Gaia likes it cool" quote. Note: In the article I was hesitating to emphasize it because exactly this remark got "expelled" from my contribution to the ECMWF newsletter (https://www.ecmwf.int/node/19356) earlier although the scientist who explained to me the data was touched himself by his observation & interpreted it this way while he explained to me his coupled system. Alas, he distanced himself from the article.

ad 2) your question: Is there a way to connect Gaia's planetary cognition more tightly to the 5-D framework of the discussion?

This question is very challenging and the answer to it surpasses momentarily the capacity of my mind since it asks for the origin of consciousness itself. Nevertheless I will try to connect Gaia's planetary cognition by finding arguments in some features of geometry of the Penrose Pattern which could eventually lead to assumptions that consciousness can be assigned to hyper-space.

---

## Referee Comment (RC3) · David Pyle (Referee) · 2 Feb 2021

This discussion paper offers an introduction to the concept and delivery of an ambitious digital science-art representation of the Earth's 'climate complexity'; Gaia 5.0. The title of the work is itself a play on themes – Gaia, representing the Lovelock and Margulis' concept of Earth as a living organism; and 5.0 variously alluding to the five-dimensional geometry of the conceptualized Earth system; but also an echo of the next generation of Web 3.0, Web 4.0 .. and so on.

The visualization itself is appealing, and sets the concept for the idea in the realms of mathematical geometry, with its use of changing graphical/wire-frame representations

around a changing planet Earth; but the clarity of the visual piece contrasts with the dense and cryptic text of the accompanying paper. The paper reads as a philosophical piece drawing on theoretical mathematical geometries and – at least to this geoscientist – is almost impenetrable, with allusions to concepts that it is assumed are familiar to the reader, rather than explained.

As an essay, I find it hard to see how this will be accessible to the diverse audience one might expect of the readership of 'geoscience communications'. I indicate a few references in the text that I struggled to understand. I am sure that the authors could address the readability/accessibility of this contribution with just a little consideration to the clarity of the introductory elements of the paper.

Concepts and statements that need a little more explanation (examples) Line 11 – 'underlying higher level space grid' Line 14 – subscendent space configuration Line 103 'pseudo-4-dimensional Minkowski-spacetime paradigm' Line 154 'lack of the human ontologic position as manifest in the absence of recognition .. etc' Line 163 'hyper-Euclidian model of the hypersphere' Line 183 'chiral movement of the 5-dimensional space' – where has the fifth dimension come from; what does it represent?

Other comments Line 33 – it seems to be a very bold claim that the SciArt project was the culmination of the 70 year old campaign? Lines 118 – 124 and 131 – 135 are repeated in the text. Line 246 'in contrast to planar geoscience models' – not sure what you are referring to here; but geoscientists (operating in deep time) have come up with many creative ways of conceptualizing at least a 4D Earth system (with time as the 4th dimension), and of representing this information in 2D visualisations. Perhaps you mean meteorological models here?

---

## Referee Comment (RC4) · Martin Archer (Referee) · 4 Feb 2021

The authors report on an ambitious and seemingly very interesting SciArt initiative which looks to represent aspects of the Earth using elaborate higher-dimensional representations. I have no doubt that this subject could be of interest to the readers of Geoscience Communication. However, in its present form I find it hard to align the manuscript with the journal's requirements (as outlined in Illingworth+, 2018, https://doi.org/10.5194/gc-1-1-2018). Principally, this states that "All research articles should include qualitative and/or quantitative evidence" which I could not see any evidence of in this manuscript. Instead, the manuscript seems to be more of an essay

or opinion piece drawing from published artistic reflections in motivating the aspects of their exhibition. I admit that I am less familiar with this style of article, but would urge the authors to consider if their paper may be better suited to a more art-focused journal where this format is more common (I am aware of the Leonardo journal where I believe articles like this are published) or to consider how it might be fundamentally revised to align with the purpose of Geoscience Communication. At present it is rather impenetrable to readers who are not fully embedded within the art world as an academic discipline. I also found the manuscript to lack a clear sense of narrative flow - sections do not naturally follow on from one another, concepts are mentioned in passing without explanation only for them to be expanded upon much later, and the overall structure and purpose of the piece is nowhere set out to readers leaving them (or least me) very confused throughout. Clearly a lot of work has gone into this project and writing it up, so I hope that my comments can be taken in the spirit of collegiality to improve the communication of this innovative work as I would truly like to understand it better!

I will summarise some of my main issues with the manuscript in its current form.

1. Scientific basis - The authors need to clearly lay out in broad and understandable terms what the scientific basis for their artwork was. Elements are dotted throughout the manuscript but are not explained to non-experts in those specific fields. Furthermore, some of the concepts seem potentially controversial, e.g. the Gaia hypothesis, but nowhere are these criticisms clearly expressed. It is rather unethical to present conjectures, not currently based in sufficient objective evidence, as fact. The potential existence and nature of the 5 dimensions behind GAIA 5.0 (and whether this is established scientific fact at present) is also incredibly unclear. While I am familiar with Minkowski 4D space-time (mentioned on line 120 but with insufficient detail for a broad readership) I could not discern what the 5 dimensions the project refers to actually are. These background introductions need to come before any discussion of the artwork is raised.

2. Clear outline of the exhibition - Trying to understand what this exhibition actually contained was also difficult to discern. Early in the manuscript aspects of it are referenced before they are even introduced. It appears that there were many different elements, could these be summarised by a diagram or table early on to indicate to readers what they were. Such a clear introduction to the artwork might help mitigate much of the confusing narrative flow.

3. Writing to the purpose of the manuscript - As mentioned previously, it was not clear to me what the manuscript is hoping to achieve. If it is to motivate the various aspects of the exhibit through previously published artistic reflections, then make this much clearer upfront to readers and outline in each section the aspect of the exhibit or the overall concept being addressed. It is not clear to me whether this would be sufficient to align with the aims of Geoscience Communication as a journal, but at least it would make the manuscript more comprehensible.

4. Objective evidence and limitations - Many bold claims about the efficacy of this project are made. However, at present they come across as purely the opinions of the authors, with no objective data gathered through a clearly defined process being presented. There is some brief mention in the Discussion section about audience experience, but this is unfortunately currently not up to the standard of Geoscience Communication being anecdotal. Limitations on what can and cannot be confidently stated based on primary research aspects to the work are nowhere to be found.

Given these rather fundamental issues with the manuscript at present, I cannot currently recommend it for publication in Geoscience Communication. I urge the authors to consider these aspects carefully to try and work out where this very interesting work would be most suitably published, how they can make it clear to readers what their exhibition entailed, and establishing to readers what the purpose of their article is. With these in mind I'm sure that the authors would be able to produce a very compelling manuscript that I would be interested to read.

---

## Referee Comment (RC5) · Martin Archer (Referee) · 7 Feb 2021

Thank you for providing links to the several interesting articles related to the Gaia hypothesis. They certainly discuss this topic in a more balanced manner, acknowledging some of the controversy and criticisms. My point was simply that the submitted manuscript does not reflect the same level of balance as these articles referenced. Therefore, I do hope you can assist the corresponding author with revisions to bring it in line with other literature as to how the Gaia hypothesis, and other scientific topics discussed, are presented. I am glad to hear that the issues of a clear objective and narrative flow are being seriously considered.

---

## Short Comment (SC1) · 7 Feb 2021

Thank you for your comment. I have been invited to participate in this artscience project by the artist Renata, thanks to my work on the Gaia hypothesis. https://www.sciencedirect.com/science/article/abs/pii/S030326472030188X https://royalsocietypublishing.org/doi/10.1098/rsif.2020.0503

My first contact with Renata has been with a complete and finished version of the draft. Since then we have had many interactions. Of the various substantial modifications to the paper that I have suggested, one of my observations was and continues to be precisely the one you mention. However, because she is the correspondent author of Printer-friendly version

the paper, I let her the freedom to accept them or not.

I agree that the paper requires direct its efforts to a clearer objective, reordering the flow of ideas and arguments and to strip away a number of unnecessary free associations that make the text impenetrable. I think this does not represent major revisions to the writing of the text, but a reordering of the content of the text so that it can be read plainly. This will be done in a new version of the manuscript that will be submitted in the next week by Renata.

However, I do not agree with your comment regarding the Gaia hypothesis. The fact that the Gaia hypothesis is controversial does not mean that it cannot be assumed in a scientific way. In general, and from my experience, I perceive that many people have a very superficial reading of the hypothesis. A more cautious review of the literature beyond Wikipedia is a good intellectual exercise. Let us remember that what we call Earth System Science has its direct origin with the Gaia hypothesis. Just to cite few you can see https://science.sciencemag.org/content/292/5524/1965 https://www.nature.com/articles/s43017-019-0005-6 https://mitpress.mit.edu/books/scientists-gaia https://mitpress.mit.edu/books/scientists-debate-gaia

Although the paper itself it is not about the Gaia hypothesis per se, nevertheless it opens the possibility from the art standpoint, that the climatic models could change their outcomes with a 5D geometric partition of the atmosphere and therefore approximate better to an integrating behavior of the Earth system as complex –living– system.

---

## Short Comment (SC2) · 11 Feb 2021

Firstly, thank you vey much for your feedback and your intention to help to improve our publication.

I am extremely grateful about being invited to publish in the GeoSciences Communications as a philosopher (art & science transfer) who works with media art and 3D digital geometry as visual tool for research and science communication.Therefor I am glad it intends to approach scientific knowledge from creativity and the arts.

My motivation is to show how can art can inspire scientific inquiry. Following your

suggestions for a better flow of ideas to make the article better readable we will use this order :

1) Present meteorologic models and the GAIA 5.0 artwork with its 3D data visualization outcome

2) Discuss Gaia mythology and the pre-Socratic ether concept – identified as geometry of the 5-dimensional space (also used in quasicrystallography)

3) Introduce 5D geometry and its possible value for the Gaia hypothesis

4) Show that 5-dimensional geometry is self-moving; here we discuss the chiral formation of cyclones and Coriolis effect and present a dynamic icosahedral aerosol formation model

5) Discuss the higher dimensional cognitive framework and its possible value for the Gaia hypothesis from a philosophical point of view.

Our attempt to apply the higher-dimensional framework on the planet Earth in order to understand its complexities is relying on earlier works concerning light as 5-dimensional electro-magnetic disturbance, quantum geometry (qubits) and for the visualizing unified theories [see ref. below]. Therefor it is our aim to outline our research about a possible application of our 5D approach to the complex system of the Earth in a way that it appears plausible and not just like an "opinion".

As supplement I attach some paragraphs from our original article which was considered to be "too scientific" by the editors. This is a reflection on science history which shows many related ideas that were integral for concepts we have today and how it can be related to a higher dimensional framework we suggest. Therefor I will put this answer to the reference list of the full short article "for further reading".

I hope this intended improvement will meet your and the Geosciences reader's expectations. Many thanks again for your effort and support as referee.

References:

Renate C.-Z.-Quehenberger: A reflection on theories of light. Quantum Theory: Reconsideration of Foundations 6 AIP Conf. Proc. 1508, 2012, 459-463; doi: 10.1063/1.4773164

RCZQ: A Proposal for a psi-ontological model based on 5-dimensional geometry, Workshop: Quantum Contextuality in Quantum Mechanics and Beyond, posterhttp://www.psych.purdue.edu/∼ehtibar/workshop/schedule.html

RCZQ:A quest for an epistemic reset in higher dimensional space, Poster Entropy Best Poster Awards, Linnaeus Conference: Towards Ultimate Quantum Theory (UQT), https://www.mdpi.com/journal/entropy/announcements, https://res.mdpi.com/data/1st-place-r.c.-z.-quehenberger.pdf

Please also note the supplement to this comment:
https://gc.copernicus.org/preprints/gc-2020-27/gc-2020-27-SC2-supplement.pdf

**Supplement:**

ÙWÚÚŠÒT ÒÞV: Investigations on science history concerning the GAIA 5.0 project.

**1.1 The question about space and motion -- from Descartes vs. Newton to Maxwell & Faraday**

According to Poincaré's philosophy, it should be assumed that there is a reality, inaccessible to us, but accessible to some being standing outside Poincaré's *cloudy planet* (Granek, 2001). The question, "La terre tourne-t-elle?", was famously posed by Henri Poincaré in a philosophical essay wherein he "doubts" the rotation of the Earth because if we want to talk about the rotation of the earth, we must first ask the fundamental question: what is motion? ,"The Earth rotates", and, "it is more convenient to assume that the Earth rotates", have one and the same meaning. Poincaré was certainly aware that the Earth moves, but he provocatively asks:

"What means do we have to distinguish absolute movement from relative movement; for example, why do we prefer the Copernican system to that of Ptolemy because it is simpler, and we conclude not only that it is more convenient, but that it is more real?" (Poincaré,1904b)

Before we introduce here the concept of an ontologic higher-dimensional space which might sound strange or even awkward to some readers, let us make briefly some historic associations:

What is remarkable about the higher-dimensional space configurations is that they must be regarded as 3D spaces in movement. This assumption goes back to the French philosopher Renée Descartes' (1596–1650) who not only invented the Cartesian coordinate system but who was the first who mentions interconnected coordinate frames for its representation. (Descartes, [1637], 1902). Descartes also re-introduced the millenia old idea of an ether into science in *Principia philosophiae* (1644). He assumed that the aether particles were in constant motion. This assumption was heretic at that time and Descartes very probably got poisoned by a Vatican missionary for North Europe because his rational ideas about higher dimensions were considered as rivaling with religion and would prevent Queen Christian of Schweden of her prospected conversion to Catholicism (Ebert, 2009). Newton felt uncomfortable mentioning the ether because he turned into Christianity and also because he had no explanation for it (Newton, 1679 Letter to Robert Boyle). His followers et drop the idea that for Newton the "absolute space" was filled with ether. Isaac Newton opposes Descartes ideas, by saying,

"If Descartes handles the translation that real movement involves not in terms of the individual particles of the vortices but in terms of the 'generic space' (his phrase) in which those vortices exist, then at last we have something we agree on. And he does say that when we are distinguishing space from bodies, we ought to understand motion in terms of space.

--- This is all wrong! Any possible motion must have a definite speed and direction." (Newton, 1699)

However, speed and direction became the core issues in modern physics while Renée Descartes' idea that all matter is just a result of transformations of spaces conforms to group theoretical descriptions of particles, celestial mechanics and Noether's theorem which proves that the symmetries of space are responsible for the conservation of energy. As Poincaré says, 5-dimensional space is appropriate for Group theory. This mans that is not only a visualisation based on fantasy but a higher-dimensional approach -- whose attempts always failed due to a lack of imagination -- should inspire to a paradigm change that can possibly help to understand a complex living systems like the Earth.

In effect, the hyper-Euclidean method we use is merely a dynamic digital vision of Euclidean geometry which Newton acknowledged as abstract reality when he said,

"So there are everywhere all kinds of figures [= 'shapes'], everywhere spheres, cubes, triangles, straight lines, everywhere circular, elliptical, parabolic and all other kinds of figures, of all different sizes, even though we don't actually see them. A physical drawing of any figure doesn't launch that figure into space; the figure was there

already, though we couldn't sensorily detect it; and the drawing is merely a corporeal representation of that already existing figure." (Newton, 1699, 9)

But since Newton not only Euclidean geometry and the ether but also space and time lost their ontologic status:

'in Newtonian physics, if we take away the dynamical entities, what remains is space and time. In relativistic physics, if we take away the dynamical entities, nothing remains. We have learned so far to give up the notions of "space and time entities" entirely since the "world is made by dynamical fields".' (Rovelli, 2006)

Recall that the notion of fields goes back to Michael Faraday (1791—1867) who used artistic methods, made drawings and experiments and developed the electro-magnetic theory which Clark C. Maxwell put into mathematical formalism. Together they were pondering the possible reality of 'lines of force' as fabric of space.

"Since at every point of space such a direction may be found, if we commence at any point and draw a line so that, [...] the resultant force at that point, this curve will indicate the direction of that force for every point through which it passes, and might be called on that account a line of force. We might in the same way draw other lines of force, till we had filled all space with curves indicating by their direction that of the force at any assigned point." (Maxwell, 1855)

These 'lines of force' may be found in our 5-dimensional space model where we use the parallels of the cubic grid (c-.f. Ammann bars, the one-dimensional version of the Penrose Pattern) where straight parallel lines into 5 directions of space can be assigned to electricity and curved lines to magnetism. From partitions of these grid of lines we can imagine -- or build in digital 3D geometry -- a discrete dynamic space similar to the phase space used in meteorology The difference to the n-dimensional phase space is that our model is composed of a 4-dimensional space in imaginary movement (C) around the 3-dimensional "compact" sphere ( $R^3$ ).

**2. Chiral dynamics and fractal characteristics of 5D space**

The flat grid model (fig. 2.) finds its equivalence in the 3D representation of the infinite 5-dimensional space (fig. 2. The concept of a discrete space as visualized by the epita-dodecahedron comprises the geometrical fact that he faces of the space cells can be enfolded into smaller and smaller regions resulting in a fractal nature of space. The logarithmic spiral is a generation method for Penrose Patterns (Kappraff, 2001). Hence both characteristics of cyclones, their fractal and chiral nature could be explained by the assumption of an underlying dynamic higher-dimensional space.

Chirality remains also an intrinsic feature of the fractal higher-dimensional dodecahedral space created by the 3D representation of the Penrose Kites & Darts Tiling. Therefor any chiral and fractal characteristics of shapes and dynamics could be traced back to this feature of space.

Passive scalar turbulence also characterizes many phenomena in the natural world, like the dramatic temperature variations between nearby points in the ocean. In that environment, the ocean currents "mix" temperatures the way stirring mixes black paint into white. Based on the continuum hypothesis that the macroscopic behavior of fluids behave as if they were perfectly continuous sine the turbulence of a small fraction will spread over the whole system, the Australian mathematician and fluid dynamicist George Batchelor (1920–2000) predicted that these patterns follow an exact, regimented order (Batchelor, [1967] 2000). This arising self-similarity is mathematically expressed by Batchelor's law which was recently mathematically proven by exploiting randomness (Bedrossian, Blumenthal, Punshon-Smith, 2019). It was suggests that in hydrodynamic turbulence each vortex contains other vortices similar to the way the nested figurines comprise a Russian doll by following an exact ratio (Hartnett, 2020).

In the context of the here presented space concept we may refer to icosahedra quasicrystals, long-range-ordered materials that lack translational invariance with 5-, 10- or 12-fold rotational symmetries. They need the same number (5-, 10- or 12-) of dimensions in order to gain a fully ordered picture. Our approach stand in the tradition of some classicists of crystallography who emphasized a connection between crystal structures and musical harmony (C.S. Weiss, J. Grassman, V. Goldschmidt, etc.). It is taking into account that Erwin Schrödinger named living bodies as "aperiodic crystal" (Schrödinger, 1948), one can suppose that biological structures are also connected with musical harmony (Stepanyan, Petoukhov et al, 2015). Hence, if we apply the same geometrical concepts which are used for quasicrystals, namely the Penrose Pattern as model for its atomic arrangement, we derive a higher-dimensional perspective on biology and the Earth as living system in the framework of a unified quasicrystallography as proposed by Alan L. Mackay (1977).

This way our hyper-Euclidean framework based on the 3D representation of the Penrose kites & darts leads us to principles of a generalized crystallography where Platonic symmetries, golden ratio and Fibonacci sequences can be observed in nature on all scales (He & Petoukhov, 2017).

**3. The Coriolis effect -- kinetic energy and the Vis Viva**

It seems notable that Coriolis was first using the term mv for kinetic energy then referred to as *forces vives* (from Latin *vis viva*, for *living force*) in his two articles on relative movements of machines (Coriolis, 1829, 1832).

In his 1829 book "Calcul de I'effet des machines" Coriolis presented mechanics in a way that could be used by the industry. He established for the first time the correct relation between potential and kinetic energy, and showed that their sum remained constant in the absence of any external force. Another 20 years later in 1853, William Rankine (1820–1872) coined the phrase 'kinetic energy' in 1853 and Lord Kelvin and P. G. Tait (1831--1901) the phrase 'potential energy' in 1862.

The vis viva was vividly discussed 100 years before between Gottfried W. Leibniz (1646--1716) and Émilie du Châtelet (1706–1749). She was not only translating Newton's *Principia Mathematica* into French and made him popular on the continent, but predicted in her theory of light what is today known as infrared radiation. She corrected Newton's assumption **mv** and squared the velocity (**v**) which resulted in the formula,  $1/2 \text{ mv}^2$  for kinetic energy. (Leibniz, 2009) Châtelet introduced the concept kinetic energy as *Vis Viva* and first proposed and tested that the total energy of an isolated system remains constant, means is conserved over time. In the 20th century the French philosopher Henri Bergson (1874–1948) referred to it as 'élan vital' in order to overcome the resistance of inert matter in the formation of living bodies. The kinetic energy (KE) — force × displacement = kinetic energy — of an object is the energy that it possesses due to its motion. But where would this energy stem from?

Earth' constant rotation is contributed to inertia. Inertia is expressed in Newton's first axiom, 'A body will remain at rest or will continue to move with constant velocity in a straight line unless acted on by an external force' is based on Aristotelian physics. The *moment of inertia* describes how difficult it is to change an object's rotational motion but it does not explain the rotational motion itself.

Despite elaborate mathematical tools which make celestial motion calculable, a philosophical demur remains: There is no celestial table from which the book which has a stored potential energy can drop to the floor. Therefor the more elegant group theoretical approach. Moreover, how much the image of Coriolis effect, -- the circle M according to Joseph Bertrand's concept (1847) which is predating Coriolis-- corresponds to the circles representing the 4D space around the Earth compact zone" with its inner core as depicted by Inge Lehmann who first determined the inne structure of the Earth (1936), here suggested can be seen here:

Figure 1 a) The tangent PM" measures the deflective angle  $2\omega$ dt which is twice the change of direction due to the rotation  $\omega$ dt. & figure inserted in the total rotational system. When this system has rotated 90° (PM' is perpendicular to M'K) the direction of the deflected motion (along PM") is parallel to its original direction (along M'K) and the moving object has followed a semi-circle, half of an "inertia circle" (Persson, 2019, 17, fig.17 b) Earth as Hypersphere with Earth's inner cores after an illustration from Inge Lehmann's 1936 paper *P*'(sketch rczq, 2020)

**3.1. The vis-viva equation and symmetries of space**

In astrodynamics, the *vis-viva equation*, also referred to as orbital-energy-invariance law, is one of the equations that model the motion of orbiting bodies. It is the direct result of the principle of conservation of mechanical energy. Noether's theorem (discovered 1915) connects the symmetry of the action of a system (the integral over time of the Lagrangian equation for the energy of a system) with conservation laws. The Lagrangian is a generalization of the formulations on constants of motion in mechanics (valid in nay dimension) which derives conserved quantities from spacial symmetries.

Actually, Noether's theorem and the conservation of energy must be linked to a 6-dimensional group of symmetries. In physics gauge symmetry or gauge theory is used to describe or "explain" natural forces in particle physics but also in celestial mechanics. — This gives rise to the *artistic* assumption that the symmetries of higher dimensional spaces act effectively as *fields of forces* that make the Newtonian description of gravity *superfluous*.

This approach would allow to regard all kinds of complex self-moving systems, such as atoms, living cells, the sun system and the universe as well as the earth as basically 6-dimensional system that relies on the geometry of space.

For the ancient Greeks the motion of the planets, light and consciousness were connected to the all-permeating aether. If we now (re-)discover higher-dimensional space configurations based on 5-dimensional geometry which Plato had in mind, the millenia old ether concept shines in a new light.

**4. A short comparison of ether concepts in the 19th century and the 5D space grid**

For Poincaré the negation of all metaphysics is still a metaphysics; and this is precisely what he called modern metaphysics (Poincaré in a letter to Flammarion, 1904). He was not convinced that Einstein found a way to eliminate the ether although he predicted that it could be declared as superfluous before him.

Relativity cannot treat action-at-a-distance and uniform rotations without retaining some kind of ether, so he changed his mind several times (Granek, 2001). Thus, the ether re-emerges as the *Je Ne Sais Quoi of Physics* (Browne, 1999). The nobel laureate Frank Wilzcek calls it a myth, repeated in many popular presentations and textbooks, that Albert

Einstein (1879–1955) swept the ether into the dustbin of history, because "renamed and thinly disguised, dominates the accepted laws of physics" and "modern quantum field theory is a direct descendant of the ether" (Wilczek, 1999).

Many artists were fond of the ether in the early twentieth century. The values of modernism was found in the complexities and contradictions of modern physics provided a fertile ground for the development of new artistic languages (Navarro, 2018). -- Mainly mute colorful comments to physics concepts which should be overcome in the 21st century by giving artists a voice (at least in SciArt publications).

Generally speaking, British scientists of the 19th century who laid the foundations of meteorology favored the existence of an all-pervasive ether. French physicists would prefer corpuscular models, while Dutch physicist H.A Lorentz as well as Maxwell were favoring non-mechanical variants of the ether. There were two competing theories, one by Augustin J. Fresnel (1788–1827) who assumed the aether was to pass freely through the Earth and one by Sir George Gabriel Stokes (1819-1903), who believed the aether was entrained by the motion of the Earth so that the velocity of the aether would be equal to that of the Earth (only near its surface). The Irish Mathematician James MacCullagh (1809-1847) contributed valuable geometrical constructions valuable geometrical constructions for the ether concept that remind us of features of the 5-dimensional space. He presented the first optical field theory in which he imagined an 'elementary parallelepiped' to be described in the ether when at rest, and then all its points to move according to the same law as the ethereal particles which compose it (MacCullagh, 1839). A parallelepiped is a three-dimensional figure formed by six parallelograms just like the subspaces of a hypercube. It corresponds also the volume to the 5D cubic grid and the model. For the same purpose, in order to find the appropriate shape of the space which enables all sorts of matter, Johannes Kepler designed rhombic  $\sqrt{5}$ - parallelepipeds, which turned out to be 4-dimensional polyhedra -- later rediscovered as Bilinski's rhombic dodecahedron -- and again re-designed as models for quasicrystallography (Levine & Steinhardt, 1986). For MacCullagh's new medium, to develop a potential function for a dynamical theory for the transmission of light is not any more a 'elastic solid' but the potential energy depends only on the rotation of the volume-elements. These volume elements remind us of the 12 respectively 12 spaces which make up the 5-dimensional space as visualized in the epita-dodecahedron

MacCullagh's equations may readily be interpreted in the electro-magnetic theory of light: e corresponds to the magnetic force,  $\mathbf{p}$  curl  $\mathbf{e}$  to the electric force, and curl e to the electric displacement, -- the same appears later in Maxwell's electrodynamics (originally relying on a mechanical-based aether) and Kelvin's theories of vortex motion, which regard rotation as analogue to magnetic force (Whittaker, 1910).

From this Lord Kelvin developed a 'Gyrostatic Adynamic Constitution for Ether' (1890) and which lead Joseph Lamor to publish a preliminary version of Lorentz transformations together with Fitzgerald's time dilation and length contraction (Lamor, 1900) which found a new interpretation in the framework of relativity theory.

This means that the principle of counter-rotation (which appears as imaginary movement  $\sqrt{-1}$ ), in the geometry of 5dimensional space configuration, in which epitahedra (E±) can be considered as counter-rotating volume-elements, can depict Lorenz transformation (as shown below). Our visualisation meets surpassingly Lord Kelvin's ideas since he also suggested as a solution for the problem of space-filling cells (the Kelvin problem) a 14-sided polyhedron (a bitruncated cubic honeycomb with curved faces and edges) based on foam experiments and dodecahedra (1887). Hence, Lord Kelvin's gyrostatic and the dodecahedral considerations are converging in the epita-dodecahedron model where the double heptahedra E± together also exhibit 14 faces. Also the famous formula E= mc2 was originally deduced from an ether model (Preston, 1875) and also deduced of John Poynting's celebrated electromagnetic Theorem by Henri Poincaré in 1900 (Auffray, 2006)

It may be an irony of history that in a mega version of Michelson's interferometer (the Laser Interferometer Gravitational-Wave Observatory LIGO) -- originally adapted to the ether-drift experiment to detect the relative motion of the Earth and the ether as proposed by Maxwell -- served meanwhile for the defection of gravity waves (Abbott et al. 2016), originally predicted by Poincaré (1905) as ultimate proof of relativity theory while ether-drift measurements and the determination of the absolute motion of the Earth was discussed but got never validated.

Notably, the first gravitational-wave signal from merging black holes was communicated by means of art: Colorful paintings and animations, including a musical score composed of converted frequency of the detected gravitational ripple, a bird-like "chirp" (Chen, 2019).

However, Poincaré's relativity theory (1895, 1902, 1905) is not concerned because his who gave the transformation the name of Lorentz-- remain valid. They describe the invariance of Maxwell's equations and must be understood in terms of symmetries of space.

However, this is what interests us here in Poincaré's 'Lorentz transformations' is the electro-magnetic field on Earth, which can be visualized in the here presented higher-dimensional model for the Earth, particularly in the 2D model of the 4-dimension embedded in 5-dimensional space (see Figure. 2. (a) The 5-dimensional space-grid and the embedding of the planet and (b) as the hyper-sphere).

**4. 2 The Phase space in meteorology -- fluctuations in the epita-dodecahedral space concept**

Figure 2 a) Epita-dodecahedron with rotated circular decoration, b) circular decoration of the blue lines only; (rczq, QC-film, Epitahedron examples, https://vimeo.com/50019507)

The phase space concept, a space in which all possible states of a system represented by a unique point was developed by Henri Poincaré, Ludwig Boltzmann & Josiah W. Gibbs. This space is full of potentially spinning points which have a physical meaning in thermodynamics and quantum physics. Therefor it is suggested that the phase space also in meteorology could not only be regarded as representation form but that its physical effects could be taken under account. A thermodynamic system consists of N particles, then a point in the 6N-dimensional phase space describes the dynamic state of every particle in that system with three-position variables and three momentum variables.

A discrete phase space is not a manifold, but admits triangulations which result in flat triangles forming simplexes. Hence the 5-dimensional unit cells used in the GAIA.5.0 film are considered as 5-simplices and therefor a viable hyper-Euclidean representation form for the 6-dimensional phase space. Especially if we consider the epita-dodecahedron as fractal entity which may be tessellated into smaller and smaller fractions of spaces we can use the circular decoration of the triangle for the representation of fluctuations that could enable the visualization of the atmospheric turbulences in detail. Chaotic fluctuations in the atmosphere could be visualized in the epita-dodecahedron model by fractals of circles in a dynamic way.

The circular waves inside the epita-dodecahedron could be developed towards a model for the simulation of fluctuations on a fractal level. In fig.18 the circular decoration of the triangles (according to the Conway decoration of the Penrose kites & darts tiling in red & blue, green circles added) is rotated and exhibits circles in motion. Here looks the epita-dodecahedron like a hypersphere. Especially the side view of rotating circles only (fig 18.b) gives an idea about the dynamic fluctuations resulting from moving points on circles. So far we can only imagine those fluctuations in a fractal manner, but a 3D animated visualization of fluctuations in a fractal epita-dodecahedral space would be a challenging art endeavor for a follow-up project.

**References**

Abbott, E.A. : Flatland: A romance of many dimensions. London: Seeley, 1884.

- Abbott B. P. et al. : (LIGO Scientific Collaboration and Virgo Collaboration) Observation of Gravitational Waves from a Binary Black Hole Merger" Phys. Rev. Lett. 116, 2016, 061102.
- Auffray, J.P.: Dual origin of E=mc, arxiv.org/pdf/physics/0608289
- Baez, J.C.: Planets in the Fourth Dimension, https://johncarlosbaez.wordpress.com/2015/03/17/ planets\_in\_the\_4th\_dimension/
- Bedrossian, J., Blumenthal, A., Punshon-Smith, S.: Almost-sure exponential mixing of passive scalars by the stochastic Navier-Stokes equations, arXiv:1905.03869v1 [math.AP] 9 May 2019.
- Bertrand, J.: Mémoire sur la théorie des mouvements relatifs", was presented by Bertrand to the Academy 21 June 1847 and published in Comptes-Rendus, 1847, 24, p. 141-42.
- Coriolis, G-G.: Mémoire sur le principe des forces vives dans les mouvements relatifs des machines, Journal de l'École polytechnique, vol. XIII, cahier XXI, 1832, 268-302 (read at the Académie des sciences, 6 June 1831 )
- ——- Mémoire sur les équations du mouvement relatif des systèmes de corps. Journal de l'École polytechnique, 24° cahier, XV, cahier XXIV, 1835, 142-154.
- Descartes, R.: Discours de la méthode [1637], Paris: Adam et Tannery, 1902.
- ------ Principia philosophiae. Amsterdam: Louis Elzevir, 1644.
- Ebert, T.: Der rätselhafte Tod des René Descartes, Aschaffenburg: Alibri, 2009.
- Göransson, J.: Symmetries of the Kepler problem. March 8, 2015. http://math.ucr.edu/home/baez/mathematical/ Goransson Kepler.pdf
- Granek, G.: Poincaré's Ether: A. Why did Poincaré retain the ether?, Apeiron, Vol. 8, No. 1, January 2001, 12.
- ----- Einstein's Ether: D. Rotational Motion of the Earth, Apeiron, Vol. 8, No. 2, April 2001, 65.
- He, M.and Petoukhov S.: Generalized crystallography, the genetic system and biochemical esthetics, Springer, Structural Chemistry, Vol. 28, No.1, 2017, 239–247

Kappraff, J.: Connections: The Geometric Bridge between Art and Science, Singapur: World Scientific Publishing, 2001.54ff.

- Kepler, J.: Mysterium Cosmographicum [1596], in: Was die Welt im Innersten zusammenhält. Antworten aus Keplers Schriften, (Ed.) Fritz Kraft, Wiesbaden: matrixverlag, 2005. 347
- —- Strena vom Sechseckigen Schnee, Neujahrsgabe oder Vom Sechseckigen Schnee, Unter Mitwirkung von M. Caspar u.F. Neuhardt, [1611] transl. by Fritz Rossmann, Berlin: W. Keiper 1943, 16 f
- Lehmann, I.: P'. In: Publications du Bureau Central Séismologique International. A14, Nr. 3, 1936, 87–115.
- Leibniz, G.W.: Sämtliche Schriften und Briefe. Band 2. Akademie Verlag, Berlin 2009, LXXXVI.
- Levine, D. & Steinhardt, P. J.: Quasicrystals. I. Definition and structure, Phys. Rev. B, Vol. 34 (2), 1986, 596 618.
- Mac Cullagh, J.: An Essay towards a Dynamical Theory of Crystalline Reflexion and Refraction, (Read December 9th, 1839.) The Transactions of the Royal Irish Academy, Vol. 21, 1846,17-50.
- McKay, A.: De Nive Quinquangula: On the Pentagonal Snowflake, Kristallogafiya 26, 1981, 910-919.
- Mackay, A. L.: Crystal Physics, Diffraction, Theoretical and General Crystallography, Acta Crystallographica Section A, Volume 33, Part 1, Jan. 1977, 212-215.
- Maxwell, J.-C.: On physical lines of force.Part II: The Theory of Molecular Vortices applied to Electric Currents, Philosophical Magazine, 21 series 4, 1861a, 348.
- ----- On Faraday's Lines of Force, Transactions on Cambridge Philosophical Society, [Read Dec. 10, 1855, and Feb. 11, 1856.] Transactions of the Cambridge Philosophical Society, Vol. X, Part I., 1864, 27-65.
- The Theory of the Primary Colours, The British Journal of Photography. August 9, 1861.

- Maxwell, J.-C.: On Faraday's Lines of Force, Transactions on Cambridge Philosophical Society, [Read Dec. 10, 1855, and Feb. 11, 1856.]
- Navarro, J. (Ed.): Ether and Modernity: The recalcitrance of an epistemic object in the early twentieth century, Oxford: Oxford University Press, 2018.
- Noether, E.: Invariante Variationsprobleme". Nachr. D. König. Gesellsch. D. Wiss. Zu Göttingen, Math-phys. Klasse 1918: 235–257,1918
- Persson, A. O.: The Coriolis Effect: Four centuries of conflict between common sense and mathematics, Part I: A history to 1885, History of Meteorology 2, 2005, 1.
- ——When Joseph Bertrand accused Coriolis of plagiarism, January 2019. http://www.bibnum.education.fr/sites/default/ files/179-bertrand\_1848-analysis-eng.pdf
- Poincaré, J.H.: A propos de la théorie de M. Larmor, L'Eclairage électrique 5 (October 5) 1895.
- ----- Lecons sur la théorie mathématique de la lumière, professées pendant le premier semestre 1887-1888, (ed.) Jules Blondin, Cours de la Faculté des sciences de Paris, Cours de physique mathématique. Paris: Carré et Naud, 1889, 1-2.
- ----- Complément à l'Analysis situs, Rendiconti del Circolo Matematico di Palermo 13, 1899, 285-343.
- ----- La Science et l'Hypothèse, Paris: Ernest Flammarion, 1902 (germ. transl. 1904).
- ---- Cinquième complément à l'analysis situs, Rendiconti del Circolo Matematico di Palermo (1884-1940) 18, 1904a, 45-110.
- ----- "La terre tourne-t-elle?," Bulletin de la Société Astronomique de France, Mai, 1904b, 216; reprinted in Jean Mawhin: La Terre tourne-t-elle ? A propos de la philosophie scientifique de Poincaré, In Le réalisme, J.F. Stoffel ed., Reminisciences 2, Louvain-la-Neuve, Paris: Blanchard, 1996, 215–252.
- ----- 3-18-2. H. Poincaré to Camille Flammarion. Archives Henri Poincaré et al., eds., Henri Poincaré Papers, Doc. 3-18-2, http://henripoincarepapers.univ-lorraine.fr/chp/text/flammarionc01.html.
- ----- Sur la dynamique de l' électron. Note de H. Poincaré. Membres de l'Académie des sciences depuis sa création; C.R. T.140 (1905) 1504-1508. https://www.academie-sciences.fr/pdf/dossiers/Poincare/Poincare\_pdf Poincare CR1905.pdf
- Preston, S.T.: Physics of the Ether, London/New York: E. & F. N. Spon, 1875
- Rovelli, C.: Philosophy and Foundations of Physics. The Ontology of Spacetime II (ChapterII: The disappearance of space and time, retrieved from https://fermatslibrary.com/s/the-disappearance-of-space-and-time) in: D. Dieks (ed. Elsevier B.V.) Elsevier, Philosophy and Foundations of Physics 4, 2006, 27ff.
- Schrödinger, E.: What is Life? The Physical Aspect of the Living Cell, London: Cambridge University Press, 1945.
- Silberstein, L.: LXXVI. Quaternionic Form of Relativity. Communicated by Dr. G. F. C. Searle, F.R.S. *Phil. Mag.* S. 6, Vol. 23, No. 137 (May 1912), 790-809.
- Stepanyan, I.; Koblyakov, A.; Petoukhov, S. The Genetic Code, the Golden Section and Genetic Music . In Proceedings of the ISIS Summit Vienna 2015, 3–7 June 2015; Vienna, Austria, T7001; doi:10.3390/isis-summit-vienna-2015-T7001
- Thomson, W. (Lord Kelvin): On gravitational oscillations of rotating water. Proc. Roy. Soc. Edinburgh, 10, 1879, 92-100.
- ----- On the Division of Space with Minimum Partitional Area, Philosophical Magazine, Vol. 24, No. 151, 1887, 503
- ----- On a Gyrostatic Adynamic Constitution for Ether" Proceedings of Royal Society Edinburgh Mar 17th 1890.
- Whittaker, E.: A history of the theories of aether and electricity : from the age of Descartes to the close of the nineteenth century, London/New York/ Bombay/ Calcutta: Longmans, Green, and Co, 1910 / Dublin: Hodges, Figgis, & Co, 1910).
- Wilczek, F.: Interview, Physics Today, January 1999.

---

## Author Comment (AC2) · 11 Feb 2021

Thank you vey much for your feedback and your intention to help to improve our publication.

Your comments & questions help us to restructure the article in order to address the readers of Geosciences journal. So we'll start with the state of the art of meteorological models and our visualization of ECMWF data. After that we will outline the higher-dimensional geometry and its application to the Earth system – inspired by models in quasi-crystallography – which has a cognitive philosophical aspect & mathematical geometrical part with a suggestion for a possible new models derived from it.

[Figure]

1) concerning Line 33 – Charles Percy Snow is usually quoted in SciArt literature, for addressing the gap between the two cultures – namely the sciences and the humanities – what he considered as a major hindrance to solving the world's problems.. What I intended to say is that it took some decades to establish SciArt as a discipline. But actually I find Bernal's lecture [1] which is addressing the ignorance arising from a lack of education more important. He addressed the weakness of scientific education, especially in England in the interwar period, which is still valid today: it offers more and more precise and progressive approximate insights into an extremely limited field and leaves out everything that defines students as people, the knowledge of Society and its history, philosophy, art, religion and morality.

2) Unfortunately, some explanations for the 'hyper-Euclidian model of the hypersphere' explaining the 4th dimension as imaginary movement of 3D spaces in contrary to the 'pseudo-4-dimensional Minkowski-spacetime paradigm' were cut out from the originally intended article due to brevity; we will explain shortly the core issues and attached below as supplement.

We intend to explain again shortly in the paper what this cognitive framework means for Earth sciences. Thank you again for your efforts as referee.

Ref.: [1] John Desmond Bernal: Science and the Humanities, lecture delivered on the 26th November 1946 at Birkbeck College, Woodchester, Stroud: Arthurs Press, Bernal, 1946)

Please also note the supplement to this comment:
https://gc.copernicus.org/preprints/gc-2020-27/gc-2020-27-AC2-supplement.pdf

**Supplement:**

Supplement:  Preliminaries of 5D geometry

Some paragraphs with a detailed description about the philosophical and geometrical approach of a 4-dimensional spacio-temporal continuum embedded into a 6D-framework.

**1. Spirals and Fractals in the geometry of the 5-dimensional space**

Spirals of galaxies and cyclones are exhibiting self-similar fractal features like the branching of rivers, veins and neurons. Similarly we find in the turbulence of air in the vortices of  taiphoons in smaller regions of air. The empirical logarithmic approximation is shown with several examples of satellite and radar images of a tropical cyclone (TC) (Yurchak, 2007). The manifold role of the golden ratio 1:1/2(1± √5) (= Phi, 1,681… or phi,0, 618…) and related logarithmic spirals have scientific significance  in science (f.e. in astronomy, biology, genetics, etc.) as well as topics of many popular „golden ratio" books. The ratio of two consecutive Fibonacci numbers (1,1, 3, 5, 8, 13, 21...) approaches the  Golden Ratio. Fibonacci series have been recently (again) raised as a principle of biological organization (Petoukhov 2016; Petoukov and He, 2010; Rapoport & Pérez, 2018).

The pattern of the golden ratio, in its 2-dimensional form as known as Penrose Tilings (Penrose, 1974) also known to represent the 2D slice of the 5-dimensional space and serves as models of quasicrystals. The 3D representation of the Penrose Kites & Darts tiling allows to visualize the infinite 5-dimensional space with the boundary of a dodecahedron. The self-similarity at every scale reminds us fractals and that ultimate chaos results in the pattern of the Golden ratio.

[Figure]

[Figure]

Figure 1 a) Generation principle of the Penrose Pattern depicted by means of a golden triangle b) Logarithmic distribution of triangles  (Source: Jay Kappraff, Connections Fig. 20.6, p. 541) c) 4-fold rotation of the 4D -Figure 2 a) representation of the Penrose kites & darts pattern, named epitahedron E+  (Quehenberger, 2014)

Hence both characteristics of cyclones, their fractal and chiral nature could be explained by the assumption of an underlying dynamic higher-dimensional space. The concept of a discrete space as visualized by the epita-dodecahedron comprises the geometrical fact that he faces of the space cells can be enfolded into smaller and smaller regions resulting in a fractal nature of space. The logarithmic spiral is a generation method for Penrose Patterns. (Kappraff, 2001)

Chirality remains also an intrinsic feature of the fractal higher-dimensional dodecahedral space created by the 3D representation of the Penrose Kites & Darts Tiling. Therefor any chiral and fractal characteristics of shapes and dynamics could be traced back to this feature of space.

However, the idea to attribute the geometrical features of 5-dimensional space in this way was not done before. Although we see  similar approaches by Descartes and Kepler  it was never recognized as such. The 5D theories Kaluza

(1921) and Klein who conceptualized it as a small circle with diameter $10^{-30}$cm and as enrolled cylindric space (1926) and de Broglie, and Pauli, several attempts by Einstein et al. are widely forgotten after a short revival in string theories.

**2  Space and time  -- the lost dimension**

The French philosopher Paul Virilio (1932--2018) dedicated a book to the »Lost Dimension« in which he diagnoses the mental perception of space in the 20th century as negation of the notion of physical dimension (Virilio,1984). In the last decades also the notions of space and time as fundamental principles were put under question.

Before we introduce here the concept of an ontologic higher-dimensional space which might sound strange or even awkward to some readers, we ave to mention the current crisis in physics and the discussion about wether space and time (or just time) are emergent (cf. Lee Smolin, Christoph Wüthrich, Carlo Rovelli et al.).

It roots in the general covariance  (point coincidence) in relativity theory, when Einstein noted, "Thereby time and space lose the last remnant of physical reality." (Einstein, 1916).

This lack of of a „real 4th dimension" seemingly results in human's cognitive lack of their ontologic position (Sloterdijk). The epistemic framework of the space where we are living is missing. This may be related to the notion „critical zone" for the biosphere. Therefor we consider it as extremely important to create a cognitive framework, that provided a space continuum.

This diagnosis of a century long erroneous development will not be corrected with one stroke. Nevertheless I hope it will flow into current research discussions.

As we all know Relativity theory lead to the merging of space and time and the introduction of relations between events in space-time. It is said that,

> 'in Newtonian physics, if we take away the dynamical entities, what remains is space and time. In relativistic physics, if we take away the dynamical entities, nothing remains.  We have learned so far to give up the notions of "space and time entities" entirely since the "world is made by dynamical fields.'(Rovelli, 2006).

Recall that the notion of fields goes back to Michael Faraday (1791—1867) who made drawings and artistic experiments and developed with Clark C. Maxwell the electro-magnetic theory. It is based on the assumption of a space filling medium, or 'lines of force' which were his fabric of space.

> "Since at every point of space such a direction may be found, if we commence at any point and draw a line so that, […] the resultant force at that point, this curve will indicate the direction of that force for every point through which it passes, and might be called on that account a line of force. We might in the same way draw other lines of force, till we had filled all space with curves indicating by their direction that of the force at any assigned point." (Maxwell, 1855)

In the 5-dimensional space model we can use the parallels of the cubic grid lines for representing those 'lines of force' of which Faraday and Maxwell were pondering if they should assume them as real or not.

We may assign straight parallel lines into 5 directions of space ( c-.f. Ammann bars, the one-dimensional version of the Penrose Pattern) to electricity and curved lines to magnetism. Hence we get a space — similar to the phase space used in meteorology — where each point in the 4-dimensional space around the 3D sphere can be assigned to an electric scalar potential and a magnetic vector potential into a single four-vector.

*Spacetime* and the problem of time was discussed since the 1960s in the context of 3-geometries used in the program of reformulation and unification called 'geometrodynamics' with the request for "first principles"(Wheeler,1967). The here presented embedding of the Hamiltonian as "first principles" into the framework of 5-dimensional space meets attempts like assigning groups of motions to a rotating 4-hyperboloid which are discussed in quantum gravity now (c.f.Hojman-Kuchar-Teitelboim, 1974). From an artist-perspective the epistemic problem can be identified as 1) a mental problem

tied to dogmatic restrictions and 2) deficits in visual comprehension, since 3-geometries remained an abstract formalism and non-Euclidean and projective geometries with pseudospheres and hyperbolic planes cannot give the full picture of a spacial continuum. Currently popular books emphasize the S3 unit sphere as model for the Earth, there is no picture for it (c.f. Unzicker, 2019).

From a phenomenological point of view we my state that we have to consider the 3D space on the sphere as belonging to the next higher order of space which is the 4th dimension. If the planet would have merely the features a 3-dimensional ball, we would be forced to believe in antipodes: Then only people on the upper hemisphere would stand upright, Australians hanging on the sphere with heads downwards etc. Only a 4-dimensional sphere creates the illusion that *every point on the sphere is on top*. (Explanation of a 6 year old child, R.Q.).

**1.1 Some geometrical preliminaries for the depiction of  time in the 4th dimension**

However, we can also think of the Earth as hypersphere analogously to a four-dimensional hypercube. A hypercube is a compound of 3D cubes on each face of a cube in the center moving in six degrees of freedom; the associated matrices correspond to the special orthogonal group of order 4, SO(4).. In the hypersphere at every point of the surface of the inner sphere a hemisphere is located. Then we my regard the space of the hemisphere in imaginary movement with the radius of the horizon as comparable with the upper half plane of the Poincaré space in the projective geometry. All those spaces in imaginary movement, depicted as counter-movement of the two small spheres in fig. 15.b (which depict the Lorentz transformation upon a Hamiltonian description) together are forming the 4-dimensional 'tangent space' of the hypersphere S3.

From a phenomenological point of view we have to experience  the 3D space, -- the biosphere which we are inhabiting together with animals and plants on Earth and in air --    as the hyperbolic space over the imaginary boundary of the horizon. Alle these spaces together on the sphere as belonging to the next higher order of space which is the 4th dimension. The loss of the horizon is a frequent topic in contemporary arts, but was already a theme of the middle ages.

**2,1  A phenomenological clarification of time and the 4th dimension**

 The  French mathematician Joseph-Louis Lagrange (1736-–1813) first mentioned the 4th dimension as principle of mechanics long ago:

"Hence, one may regard mechanics itself as a geometry of the 4th dimension." (Lagrange, 1797, p. 223)

This fact is not so commonly known, because the Lagrangian function uses generalized coordinates for generating the correct equations of motion which summarizes the dynamics of the entire system and characterizes the possible motion of the particles also applied to meteorology.

The 19th century was dedicated to exploring the 4th dimension. A phenomenological explanation of the 4th dimension was first given in 1846 by the German philosopher, physicist and experimental psychologist Gustav T. Fechner's (1801–1887) in his ironical, funny philosophical essay *The 4th dimension* (Fechner, 1846). There he describes the imaginary movement of a row of houses ( fig.2. b) as countermovement to the rotating Earth which brings us every instant in a new space while we consider ourselves as being in rest. That's why we need the 4th dimension !

[Figure]

[Figure]

Figure 2 a)  4D Hypercube with house   b) illustration of the 4th dimension, a row of houses changing space over time after G.T. Fechner, (still frames from "Passage to Level-1" , R.Q, QC: vimeo.com/39339348 )

Fechner's "shadow-beeings" inspired Edwin Abbott Abbott (1838--1926) to his influential mathematical satire «Flatland -- A Romance of Many Dimensions« (1884 ) in order to illustrate that mathematicians are hardly able to imagine a sphere, not to speak of any higher dimensional object. »Flatland« is still referred to as "proof" of a "natural   in-imaginability of higher dimensions" in mathematics books, -- despite his plea,

> "Yet I exist in the hope that these memoirs ... may find their way to the minds of humanity in Some Dimension, and may stir up a race of rebels who shall refuse to be confined to limited Dimensionality." (Abbott, 1884)

Around the same time, amidst popular discussions about the 4th dimension, Alicia Boole-Stott (1860-1940), daughter of the mathematicians Mary Everest and George Boole, the *The Princess of Polytopia*, who later conveyed her knowledge to the young H.S.M.Conway, first visualized all 4-dimensional Platonic solids (Boole-Stott, 1900). Her friend H. G. Wells should then *invent* a »Time Machine«, in order to travel on the time-axis towards a distant future (Wells, 1895). Hence, the debate about time travel is still a serious issue in physics while William Rowan Hamilton' (1805--1865) mathematical foundations for it in the »Algebra as the Science of Pure Time« (Hamilton, 1837) seem to be wildly forgotten. Together with the writer and poet Edgar Allen Poe (1809--1849) who proclaimed the expansion of the universe science-critical cosmogony »Eureka « (1848)   -- which Einstein was reading in 1933 and 1940 before he agreed to Lemaître's idea of an expanding Universe -- these three artists probably produced the most eminent influence on science of the last 120 years (van Slooten, 2017). -- What about vice versa?

In his »Lectures on Quaternions« (1853) Hamilton noted that complex numbers (quaternions) are a composite of real and imaginary numbers ($z = x + iy \in C$ where $i = \sqrt{-1}$) living in four-dimensional space, --  a piece of the *curved space* surrounding a sphere. Therefore time and the 4th dimension is intrinsically connected to the surface of a sphere,--

Although distinct from the relativistic curved--space-time-concept following from Minkowski's "pseudo-4-dimensional" geometry in which realized Einstein dream of traveling on a beam of light, the principle of reality can be expressed in a quaternionic way (Silberstein, 1912).

[Figure]

[Figure]

Figure 3. a) Image: Real time DGPS (GPS Tutor, 1998), https://nptel.ac.in) b) The Lorentz transformation embedded in the 4D space; A and B in the grid of the Ammann bars, the discrete space elements for a dynamic representation of the factor √-1, z = iy, (sketch RQ 2013)

By relying on Hamilton, the most important characteristics of a 4-dimensional system is the imaginary factor √-1 which is sometimes hidden or left out. F.e. Stokes shows how to omit √-1 in his equations on frictions of fluids (Stokes,1850, 15). Lord Kelvin needs to put 'properly the imaginary constituents' on the equations of the *Kelvin waves* (Thomson, 1879). If we compare the GPS set-up (3.a) with the quaternionic description (3.b) it is easy to see the geometrical difference of the technological realization of Minkowski's principle to the Hamilton's quaternionic principle of time. This conforms also to Robert Rosen's call for the requirement of a new concept:

> 'The surface of the sphere is in some sense a limit of its planar approximations, but to specify it in this way **requires a new concept (the topology of the sphere)** that cannot be inferred from local planar maps alone.' (Rosen, 1985)

We suggest the topology of the hypersphere relying on Silberstein's quaternionic formulation of relativity. As Conway notes in his letter to the editors attached to the Silberstein paper,

> „Beyond this, however, the quaternion has the advantage of being asymmetrical, the time-scalar occupying a different position from the space-vector. It is thus more in touch with real phenomena." (Conway's comment to Silberstein,1912)

However, we can also think of the Earth as hypersphere analogously to the four-dimensional hypercube. A hypercube is a compound of 3D cubes on each face of a cube in the center moving with degrees of freedom. In the hypersphere at every point of the surface of the inner sphere a hemisphere is located. Then we my regard the space of the hemisphere in imaginary movement with the radius of the horizon as comparable with the upper half plane of the Poincaré space in the projective geometry. All those spaces in imaginary movement, depicted as counter-movement of the two small spheres in fig. 3.b (which depict the Lorentz transformation upon a Hamiltonian description) together are forming the 4-dimensional 'tangent space' of the hypersphere $S^3$.

**References**

Abbott, E.A.: Flatland: A romance of many dimensions. London: Seeley, 1884.
Boole - Stott, A.: On certain series of sections of the regular four-dimensional hypersolids, Verhandelingen der Koninklijke Akademie van Wetenschappen te Amsterdam (eerste sectie) 1900.
Einstein, A.: Die Grundlagen der allgemeinen Relativitätstheorie. Annalen der Physik 49, 1916, 769–822
Hamilton, W. R.: Theory of conjugate functions, or algebraic couples, with a preliminary and elementary essay on algebra as the science of pure time [1835] Transactions of the Royal Irish Academy, Vol. 17, part 1, 1837, 293 -422. In: H Halberstram, R.E Ingram (ed.): Mathematical papers of W. R. Hamilton, Vol. III, New York / London: Cambridge Univ. Press, 1967.
—— Lectures on quaternions : containing a systematic statement of a new mathematical method, of which the principles were communicated in 1843 to the Royal Irish academy, and which has since formed the subject of successive courses of lectures, delivered in 1848 and subsequent years, in the halls of Trinity college.Dublin: Hodges and Smith, 1858
Hojman, A.S, Kuchař, K., Teitelboim, C.: Geometrodynamics Regained, Annals of Physiks 96,1976, 88-135.
Poe, E.A.: Eureka, A Prose Poem: Or the Physical and Metaphysical Universe. New York: Leavitt, Trow & CO. Prs., 1848.
Kappraff, J.: Connections: The Geometric Bridge between Art and Science, Singapur: World Scientific Publishing, 2001.54ff.

Maxwell, J.-C.: On Faraday's Lines of Force, Transactions on Cambridge Philosophical Society, [Read Dec. 10, 1855, and Feb. 11, 1856.] Transactions of the Cambridge Philosophical Society, Vol. X, Part I., 1864, 27-65.

Penrose, R.: The role of aestetics in pure and applied mathematical research. Bull.Inst. Math.Appl. 10, 1974, 266-271

Petoukhov, S. and He, M.: Symmetrical Analysis Techniques for Genetic Systems and Bioinformatics  Theory, Methods and Applications, Somerset : Wiley, 2011.

Rovelli, C.: Philosophy and Foundations of Physics.The Ontology of Spacetime II ( ChapterII: The disappearance of space and time, retrieved from https://fermatslibrary.com/s/the-disappearance-of-space-and-time) in: D. Dieks (ed. Elsevier B.V.) Elsevier, Philosophy and Foundations of Physics 4, 2006, 27ff.

Rapoport, D.L.,   and   Pérez, J.C.: Golden ratio and Klein bottle Logophysics: the Keys of the Codes of Life and Cognition, Quantum Biosystems,  9 :2, 2018, 8-76.

Silberstein, L.: LXXVI. Quaternionic Form of Relativity. Communicated by Dr. G. F. C. Searle, F.R.S. *Phil. Mag.* S. 6, Vol. 23, No. 137 (May 1912), 790-809.

Stokes, G.G.: On the effect of the internal friction of fluids on the motion of pendulums by Stokes [Read December 9, 1850.] [From the Transactions of the Cambridge Philosophical Society, Vol. IX; Reprinted in Mathematical and Physical Papers, Sir George Gabriel Stokes and Sir J. Larmor, Vol. 3, 1880-1905]

Unzicker, A.: Die mathematische Realität: Warum Raum und Zeit eine Illusion sind, (Mathematical reality: Why space and time are an illusion, Selbstverlag, 2019, 144-207.

Wells, H.G.: The Time Machine: An Invention, London: Heinemann, 1895.

Yurchak, B.S. Description of cloud-rain bands in a tropical cyclone by a hyperbolic-logarithmic spiral. Russ. Meteorol. Hydrol. 32, 2007, 8–18. doi.org/10.3103/S1068373907010025

---

## Short Comment (SC3) · 17 Feb 2021

Dear Prof. Clarke,

Thank you very much for your comment and your very provocative question.

Personally I do not see that the 5-dimension, hyper-geometrical dimensionality and even fractals by themselves support a sort of planetary sentience, or sensitivity beyond metaphorical free associations. This is because two sort of arguments:

1- only the realization of self-referential (autopoietic) systems is able to enaction and sense making. The Gaia 5.0 project is not about describing the form of autopoietic

systems or to show the causal relations that allowed us to pass from 5D geometry to sense-making.

2- if there is some kind of geometry that must be link to the self-referential systems it should be the one I call the Strong-geometric entropy (a) and the Soft-information geometry (b).

a) the geometric entropy is relate to the resolution of Poincare's conjecture https://science.sciencemag.org/content/314/5807/1848 in the geometric structure of the Ricci flow (https://arxiv.org/abs/math/0211159, https://arxiv.org/abs/math/0303109, https://arxiv.org/abs/math/0307245) and its possible connection with the thermodynamics of black holes, which the art-science Gaia 5.0 project does not intend to show.

b) the information geometry is related to the formal expression of autopoietic systems and their boundary in dynamical and Makovian terms https://royalsocietypublishing.org/doi/full/10.1098/rsta.2019.0159. This means that any autopoietic system, including the earth system https://royalsocietypublishing.org/doi/10.1098/rsif.2020.0503, equipped with a Markov boundary may have sensory activity and therefore sentience https://www.mdpi.com/1099-4300/22/5/516. The Gaia 5.0 project, in its current version, is not linked to this.

The art-science Gaia 5.0 project rather propose a 5-Dimentional partition of geometrical space of the atmosphere instead of the cubic 4D partition that the current climate models are based on. Therefore, this art-media proposal suggests that 5D partition could allow us to 'see' perhaps dynamics of the atmosphere far beyond current climatic models based cubic 4D, and thus a closer approximation to a living –Gaian– system. However, this should be clarified in a further version of the manuscript.
* * *

---

## Short Comment (SC4) · 18 Feb 2021

Dear Prof. Clarke,

"Thank you for encouraging us to emphasize the aspect of Gaia as a 'cognitive system." ... sorry for my bad English, "I am Austrian" and sometimes Germanicisms lead astray.

I appreciate your hint to your article on "cognition not consciousness": Following Margulis' and your thoughts on Gaia's planetary cognition manifested by the Earth system and its self-regulatory loops we may well re-consider the intelligent subscendent pattern that creates cognition that occurs both above and below the level of thought. It allows for systemic responses beyond awareness and without consciousness (Bruce Clarke, Cognition not Consciousness, February 26, 2019)

Thank you for giving us the hint to the cybernetics notion "auto-poesis": My geometrical research was inspired a lot by the "pattern that connects", Gregory Bateson's concept which I learned from the charming Austrian co-founder of cybernetics, Heinz von Förster. When Heinz von Foerster spoke of "self-organizing demons which govern the intrinsic structural and material energetic properties of the building blocks" of living systems he imagined magnetic cubes reminiscent of Charles Howard Hinton's tesseracts or hypercubes, drawn as "Jumbled Boxes" by Gordon Pask (Clarke 2009, p.50) Of course motion and the idea of dynamic building blocks can only be visualized by means of 3D animated geometry. Instead of 4D hypercubes we use the 5-dimensional cubic grid, and its dual the structure of the infinite 5-dimensional space in the shape of the epita-dodecahedron which would then serve as "auto-poetic" subscendent "imaginary machine".

From this 5-, respectively 10-dimensional space where quasicrystals "live" (Shechtman et al 1984), we can easily draw a connection to Schrödinger who compared life with an "aperiodic crystal"and his note about a living cell being 4-dimensional (Schrödinger, 1944). Our hyper-Euclidean geometry is inspired from this hint and forms 4-dimensional units for the Earth and living cells. Based on the "golden" grid (âĎď/5) it builds a "real" 4D tangent space not from a point but from a 3-dimensional sphere in the center ( cf. Bolyai's non-Euclidean geometry). Rashevsky also showed that Riemannian geometry is conforming to Golden ratio (Rashevsky, 1967). This geometrical approach allows to draw a model of 4-dimensional living cells. It allows to visualize symmetries and group transformations in a spacial continuum. Distinct to projective spaces hyper-Euclidean geometry is adding infinitely many spaces R5 inside R4. So we get a closed structure with boundaries for R3, R4 and R5 inside R6 . Fig 1. (a) shows such a 4D model of a cell and its symmetries (b) and organelles of an eukatiotic

**GCD**
cell are composed of organelles, including mitochondria (cellular energy exchangers) in the shape of a Bucky-ball. This way our hyper-Euclidean framework based leads us to principles of a generalized crystallography where Platonic symmetries, golden ratio and Fibonacci sequences can be observed in nature on all scales (He & Petoukhov, 2017). Hence gradients in the sense of Gaia as a "gradient-reducer" (Schneider and Sagan 2005) concerning minimizing free energy (Rubin 2019, 2021b) might become a different meaning in the 4-dimensional framework: Instead of independently moving particles driven by thermodynamics, we would get Schrödinger QM relations between the 4-momentum and the 4-gradient  $\partial$ . If the quantum relation is applied to a 4-vector field A $\mu$  instead of a Lorentz scalar field then one gets the generalized electric potential and a generalized magnetic potential. It would be interesting to find out, if it makes sense to apply the free Maxwell equation – when the rest mass term is set to zero (light-like particles) to explain a living cell.

– Alas, your question about the "sentient" and cognition" will lead us then to the problematic realm of the 5th dimension – which de Broglie recognized as field of matter waves (1927) and about which Einstein asked for a physical explanation. But we will try to address some points in our article.

According to our 6-dimensional parameter-space model (Fig.2) e may imagine that R5 is the space which produces not only light (Quehenberger, 2012) but micro-wave phenomena – previously called "spirit" – would lead us to Norbert Wiener's "frequency domain" and to the Pribram & Bohm theory of consciousness. We can imagine these frequencies tied to fractions of space, like the Conway circle decoration of the Penrose pattern is fixed on golden triangles forming waves in motion. Since Penrose patterns are perfectly aperiodic but are mirroring themselves at infinity, the term "reflection" gains a geometrical form. One might suppose that a similar recursive effect could take place at the tiniest triangular fractions of space that mirror themselves in a chiral movement towards a "zero point at infinity" (probably inside a 7-dimensional neuron?) and flash back in the emergence of a faction of recognition. The connection between conscious-
ness the Unified Field and ancient philosophies was made before (c.f Hagelin,1987) F.e. chaos theorist Ralph Abraham proposed a mathematical quantum vacuum model of consciousness expanded to the self-organization of the gross ether from a subtle (submicroscopic) cellular network (Abraham and Roy, 2010). Based on Bohm's theory Rupert Sheldrake developed his morphogenetic field theory in order to explain processes in living systems, plant growth (but remains agnostic about an underlying structure).

Much more research is necessary to confirm my artistic speculations and eventually the scientific paradigm would have to change so that we can define how "consciousness" could arise from unitary structures. If so we can Gaia even call a hyperconsciousness,— like our paleolithic mothers probably have thought of it when they've invented the Great Goddess: ""Earth on a global scale—all these things resonate strongly with the ancient magico-religious sentiment that all is one" (Margulis & Sagan, 2011). At least, with the Platonic assumption that "the pattern itself is an eternal being" , like Plutarch speaks only of the "animated movement" but not of a singular "soul", we get rid of the mind matter problem.

**References**

Abraham, R. and Roy, S. (2010) Demystifying the Akasha: Consciousness and the Quantum Vacuum, Epigraph.

Clarke, B. (2019) Cognition not Consciousness —for the panel "Scale" at the Crafting the Long Tomorrow conference, Biosphere 2, February 21-24, 2019

Clarke, B (2009) Heinz von Foerster's demons, in: Bruce Clarke, Mark B. N. Hansen: Emergence and Embodiment: New Essays on Second-Order Systems Theory, Duke University Press, pp.48-49

Clarke, B., (2020a) Gaian Systems: Lynn Margulis, Neocybernetics, and the End of the Anthropocene. University of Minnesota Press, Minneapolis.

**GCD**
de Broglie, L. (1927) "L'univers à cinq dimensions et la mécanique ondulatoire," Journal de Physique et le Radium, vol. 8, 66-73.

Margulis,L., Dorion Sagan, D. (2011) : Dazzle Gradually: Reflections on the Nature of Nature, Vermont: Chelsea Green, 2011

Rashevsky, P. K.(1967) Riemannian geometry and tensor analysis. Nauka, Moscow

Rubin, S., (2017) From the cellular standpoint: is DNA sequence genetic 'information'? Biosemiotics 10, 247–264. https://doi.org/10.1007/s12304-017-9303-x.

Rubin, S., Veloz, T., & Maldonado, P. (2020). Beyond planetary-scale feedback self-regulation: Gaia as an autopoietic system.ÂăBiosystems, 104314.

Hagelin, J. S. (1987). "Is Consciousness the Unified Field? A Field Theorist's Perspective", Modern Science and Vedic Science, 1, pp. 29–87.

He, M. and Petoukhov, S. (2017) Generalized crystallography, the genetic system and biochemical esthetics, Springer, Structural Chemistry, Volume 28, IssueÂă1

Schneider, E. D., & Sagan, D. (2005). Into the cool: Energy flow, thermodynamics, and life. University of Chicago Press.

Schrödinger, E. (1944) Was ist Leben? - Die lebende Zelle mit den Augen des Physikers betrachtet, engl. origin. 1944, Piper, Munich 2010

Sheldrake M. & Sheldrake R.(2017) Determinants of Faraday Wave-Patterns in Water Samples Oscillated Vertically at a Range of Frequencies from 50-200 Hz, Water (2017) Vol 9

Quehenberger, RCZ, Skepper, B. Stepanyan, I.V. (2020) [C G; A T] Epita Matrix Genetics - Towards a Visualization of Genetic Codes via Genetic Music, Proceedings of Cairotronica 2016, Leonardo Vol. 53:1, Feb, 2020

Quehenberger, RCZ, (2012) A reflection on theories of light", Quantum Theory: Recon-
sideration of Foundations 6 (Eds:) Andrei Khrennikov, H. Atmanspacher, Alan Migdall, and Sergey Polyakov, AIP Conf. Proc. 1508, pp. 459-463

**GCD**
GCD
**Fig. 1.** 4D framework of a cell with "real" tangent space (R4) with inherent icosahedral symmetries in R5 (small picture in grey) around the center of the cell (R3) image: RCZQ (b) Image of a

---

## Editor Comment (EC1) · Tiziana Lanza (Editor) · 22 Feb 2021

Dear All (reviewers and authors),

first of all, thank you very much for engaging in this complicated peer review. I am the editor handling this paper, and I feel I have also to intervene in the discussion, in the hope that the article will be finally shaped in the right way to be published.

1: I would like to remind once again Renate and co-authors to carefully read what M. Archer has also suggested: Illingworth+, 2018, https://doi.org/10.5194/gc-1-1-2018 . This is extremely important.

[Figure]

2: Motivation and target. I had the pleasure to visit the installation in two occasions: at the Joint Research Centre (in a scientific context) and at Bozar (artistic context). For this reason, I have asked Renate to clearly state what are the motivations behind this SciArt installation. As far as I have understood the authors would like to raise a debate inside the scientific community. Is this true? Can I ask the authors to clearly state their motivations behind this SciArt installation? Please think about it!

This should be clarified also when they write about the target. The main target is the scientific community? In this case, the present work can be an interesting example on how art can be used to raise a scientific debate. Then, there is another level through which, by the means of art the general public is introduced in what it is described in 94-100. The voice of Lydia Lunch, the creepy music instil in the visitors' feelings of respect towards Gaia, the Mother Earth. Can you please let us know clearly what kind of message you intend to address to the general public?

3: please have in mind that the subject of the present article (to be included in the present Special Issue) is the SciArt installation, as I have told Renate several time. Please be sure that all the people that haven't visited the installation can have at least an idea of it. How can you do this:

Describe early in the paper how this installation works clearly. Paragraph 2 (GAIA 5.0: The installation as it is shaped now doesn't do it). Immediately put the link regarding the installation, the video supplement: https://av.tib.eu/media/49792. Make immediately reference to the text performed by Lydia Lunch. Include also and immediately the timeline of the video. Add further details as: what the visitor of the installation is supposed to do. Give also some details about the environment of the installation. Where the visitor is supposed to watch the video? In a TV screen? Projected in the wall. . . And the sound is loud? What is the role of the text performed by Lydia Lunch? Is the visitor supposed to pay attention to the words performed, or the performance of Lydia Lunch is just evocative?

4: Since the text provided in the appendix immediately introduces the reader in the scientific debate raised, now it comes the time to dedicate a well shaped paragraph to the philosophical and scientific background of the SciArt installation. The paragraph can be built taking some key concepts in the text already provided in the appendix as: "infinite 5-dimensional space" "Pangaea the dreaming creature" the reinstated concept of "aether" "the new interpretation of the earth' intrinsic dynamic relying on the geometry of space itself", and so on. Please do it in plain language avoiding as much as possible jargon, as it is very well suggested by D.Pyle, but also by M.Archer, both declaring the article being impenetrable and cryptic.

5: How do you decide if your SciArt installation has accomplished what you intended to in conceiving it? There is no evaluation in the present paper but at least let us know about your future plans. Do you intend to reinstall it in other context? Are you planning to investigate the effects of the installation on the different kind of public? Do you intend to investigate the effects of your installation inside the scientific community? I believe this would be interesting. . .

---

## Author Comment (AC3) · 21 Jul 2021

**Dear reviewers,**

many thanks for your time and consideration also on behalf of all co-authors. We are grateful for your valuable inputs and suggestions to improve our publication. Please allow me a final remark concerning the problem our co-author Sergio Rubin brought up and which might also be of common concern since we use the term "Poincaré homology sphere" in our Gaia 5.0 video and publication:

Indeed, our geometrical concept is not linked to Perelman's work which commonly appreciated as proof of the Poincaré conjecture. We are strictly relying on Poincaré's idea's, his search for the fundamental polyhedron which we assume to having identified as 3D representation of the Penrose Kites & darts tiling ( $E\pm$ ), the unit cell of the 5-dimensional space, - the space which Poincaré himself found appropriate for group theory (Poincaré, 1899, p.19). If we look at attached images of the visualisation of the epita-dodecahedron, - a simply connected (?) manifold of golden triangles forming the shape of a 4-dimensional dodecahedron where the blue circle decoration (fig.1) is forming elliptic curves in motion (fig.2). It reminds us very much on Poincaré's sentence: "The analogy with the formation of cycles in the theory of Fuchsian groups is evident. It is even stronger if we assume there is only a single polyhedron P1 ." - This is also how we may imagine the one-dimensional linear flow attached to the polyhedral space, where the finite group of rotations SO(4) acts freely in the tangent (beyond the red sphere, while the inner space belongs to S2) on S3. The picture of the dodecahedral space with circles in motion is clearly matching our picture of the hypersphere which we use as a model for the Earth is built from icosahedral symmetries.- the dual of the dodecahedron (fig.3). Distinct to Poincaré's approach Perelman's work is based on Hamilton's Ricci flow method that acts on a 3-manifold with a Riemannian metric. For this he introduced celebrated new algebraic tools in differential geometry that found many applications on dynamical systems since then. While Perelman's work did give convincing arguments for eliminating Hamilton's problem with the cigar soliton, he did not claim to have proven the Thurston Geometrization Conjecture - which was an attempt to prove the Poincaré conjecture - until the end of a second even more difficult paper, "Ricci Flow with Surgery on Three Manifolds" (2003) but he never claimed for what he is credited nor did he ever refer to Poincaré's original problem in any of his three seminal papers (Perelman, 2002, 2003a b). As John Stillwell (2009) emphasized, Poincaré's analysis situs treatises are very different from what we know as Perelman's solution to the Poincaré conjecture that involves physical properties such as time, entropy and heat.

Although there is a common ground between Ricci and Poincaré their works are very
different: Poincaré is clearly building on his experience with Fuchsian groups in the early 1880s (see Poincaré (1985) where each group is associated with closed paths in a 3-manifold defined by a polyhedral region with faces identified by certain geometric transformations (Poincaré, 1892). Also Gregorio Ricci was influenced by Fuchs when he wrote a thesis on differential equations, entitled "On Fuches's Research Concerning Linear Differential Equations" With Tullio Levi-Civita he worked on the "absolute differential calculus" with coordinates or tensor calculus on Riemannian manifolds. But both work on very different approaches: Ricci presented a system of point transformations in order to determine covariant functions in differential geometry while Poincaré's mathematical edifice is clearly built on higher dimensional spacial formations, dedicated to group theory permutations of polygons and polyhedra and the search for the fundamental polyhedron.

Our construction of the hypersphere is very much conform with the horosphere, a Euclidean sphere tangent to the unit sphere where the point X of tangency is the center of the horosphere. Also his "ideal, hyperbolic simplex" with dihedral angles  $60^{\circ}$  of which he takes two copies and glues the faces together is conform to one side of our polyhedra  $E\pm$  face we suggest. Furthermore  $E\pm$  is exhibiting the Poincaré wished.

Alsing, Miller, T.-S.Yau et al, described the Ricci Flow of a piecewise-flat simplicial geometry. Their model introduces piecewise-flat 3-geometries considered here are built of isosceles-triangle-based frustum blocks (solids that lie between one or two parallel planes cutting it) with regular icosahedra as S2 cross-sectional geometries (fig.4). They remind us of our irregular pyramids  $E\pm$  which are whose edges are composed of parallels of the grid Z5 and can be composed to icosahedra (as shown in the video Gaia 5.0 as model for the formation of aerosol).

Quite a few more arguments (for which here is no space) would be available in support of our claim as well as even more open questions concerning our visualisation (after the description of Threlfall and Seifert, 1933 with twists of opposite lying faces but not by  $3\pi/5$  radians but by  $\pi/5$  radians as in the case of spherical dodecahedron space).
We hope we will able to discuss it one day with the mathematical community in order to justify our use of the term "Poincaré homology sphere" in the here discussed paper as model for the Earth complex dynamic system and elsewhere.

References: P.M. Alsing, W. A. Miller, M. Corne, X. Gu, S. Lloyd, S.Ray, S-T. Yau (2013) Simplicial Ricci Flow: An Example of a Neck Pinch Singularity in 3D, arXiv:1308.4148v2.

Fuchs, L. (1880) Über eine Klasse von Funktionen mehrerer Variablen, welche durch Umkehrung der Integrale von Lösungen der linearen Differentialgleichungen mit rationalen Coeffizienten entstehen", J. Reine Angew. Math., 89: 151–169

R. S. Hamilton (1993) The formation of singularities in the Ricci flow, in Surveys in differential geometry, Vol. II and Internat. Press, Cambridge, MA, 1995, 7–136.

G. Perelman (2002) The entropy formula for the Ricci flow and its geometric applications, arXiv: math.DG/0211159v1. G. Perelman (2003a) Ricci flow with surgery on three-manifolds, arXiv: math.DG/0303109 G. Perelman (2003b) Finite extinction time for the solutions to the Ricci flow on certain threemanifolds, arXiv: math.DG/0307245.

Poincaré, H. (1899) Supplement to Analysis Situs, Rendiconti del Circulo Matematico di Palermo 13, 1899, 285-343. transl. by John Stillwell: Papers on Topology, Analysis Situs and Its Five Supplements, Henri Poincaré, 2009.

W. Threlfall and H.Seifert (1933) Topologische Untersuchung der Diskontinuitätsbereiche endlicher Bewegungsgruppen des dreidimensionalen sphärischen Raumes, Mathematische Annalen 104 ,Volume 107, No.Âă1, 543–586

W.P. Thurston (2002) The Geometry and Topology of 3-Manifolds, Electronic version, http://www.msri.org/publications/books/gt3m/
Fig. 1. Epita-dodecahedron with blue horocycles in motion flow (RCZQ, QC, 2012)